# Inferring Spatial Distance Rankings with Partial Knowledge on Routing Networks

**Dominik Köppl** 🔵

M&D Data Science Center, Tokyo Medical and Dental University, Tokyo 113-8510, Japan; koeppl.dsc@tmd.ac.jp; Tel.: +81-3-5280-8626

**Abstract:** The most common problem on routing networks is to compute the shortest paths from a source vertex to a set of target vertices. A variation of it, with applications for recommender systems, asks to merely rank the target vertices with respect to the shortest distances from the source. A classic solution is Dijkstra's algorithm; however, it is too slow for large but meaningful applications. A setting where the target vertices are fixed but the source vertex is only known at query time allows for preprocessing. Following the line of research on preprocessing the routing network to speed up the computation of shortest paths, we study in this article a novel approach tackling the problem of ranking the static set of target vertices on the routing network with regard to the distance from a given source vertex to these target vertices by leveraging preprocessing. Our approach allows us to generate a partial solution by pre-computing the distances between all targets so that a shortest-path algorithm does not have to determine the shortest path from the source to every target in general. Our proposal can be adopted for both static and time-dependent networks, and it can be used in conjunction with a general shortest-path algorithm. We can experimentally observe significant speed-ups when using our proposed techniques.

**Keywords:** routing network; quasimetrics; time-dependence; caching

## 1. Introduction

The proliferation of geo-spatial data such as OpenStreetMap (OSM) brought new ways for reasoning about spatial properties. Since the advent of complex mobile devices, spatial query problems emerge in ubiquitous application cases of common life. The most common question about the comparison of several distant points is "which point is closer to one's current location?". Hence, an answer does not need to provide exact numerical computation of the distances, but a ranking of these points. For illustration, let us consider a tourist searching for a nearby restaurant using their Global Navigation Satellite System (GNSS)-enabled mobile device (see [1,2] for multi-criteria use cases). A recommender system will prefer those Points Of Interest (POIs) that are in the neighborhood of the query point but also fulfill certain properties appreciated by the user. Using the Euclidean distance as distance can provide useful results in most cases. Unfortunately, one-way roads, rivers traversable by certain bridges, and complex landscape structures such as archipelagos will likely cause unexpected results. As a matter of fact, we dismiss the Euclidean metric and use the existing routing network for computing distances. See [3] as an example for the topological analysis of existing public transport system routing networks.

Let us take into consideration that routing networks are not necessarily undirected: One of the most obviously directed networks is Bolivia's zip line system used for the transportation of laborers and harvest goods. Zip lines are solely propelled by gravity and the momentum of the user. Therefore, a return journey often means to take a detour, e.g., climbing a hill in order to pass a gorge. Furthermore, let us put the notion of the length of a path into a more abstract sense: Fuel or calorie consumption and estimated time are valid examples for path lengths that will render a graph directed. Again, with the altitude

in mind, it is easy to see that downhill driving will cost less energy than uphill. See also [4] for a use case of the energy consumption of electric vehicles.

Alas, routing on networks without pre-computation is tedious and takes an unsatisfying amount of time with dense network data and distant targets. Therefore, we pre-calculate some distances on a given dataset (here restaurants) in order to partially estimate for any query point the weak ordering between the targets induced by the question: which POIs are closer to the query point? We take the standard assumption that pre-computation of the distances between all vertices is too costly. Hence, we stipulate that the vast majority of vertices are not targets.

A concrete example of the problem is given by skyline computation: Let us assume that our tourist adds multiple attributes to their query: e.g., price, type of restaurant, and so forth [5]. As a tourist usually wants to have a good compromise of all attributes to choose from, the system will compute the skyline of all restaurants in this area. This set consists of all restaurants that are not worse than any other restaurant with respect to all attributes. While most papers focus on the Euclidean distance as a metric for spatial attributes [6], there have already been a few demonstrations that combine routing network search with skyline computation such as [1] or [7]. Coupling multiple ranking queries on routing networks will obviously lead to excessive computation load when neglecting indexing or caching. Although many skyline algorithms such as skyline breaker [8] involve the explicit evaluation of scoring functions (e.g., an enumeration of our ranking), a simple algorithm such as block-nested loop [9] that merely uses pairwise tuple comparison could be used along with our proposed method to speed up skyline computation.

## 2. Related Work

As routing network data such as OSM spans huge graphs, slack performance is a well-known problem of geolocation services that employ a routing network for measuring distances. Without any prior knowledge and assumptions on the network, Dijkstra's algorithm [10] implemented with Fibonacci heap is the (asymptotically) fastest known solution [11].

For exact search, the golden rule is to prune data [12] as soon as possible in order to avoid unnecessary evaluations. Goldberg [13] presented a survey article about the most common vertex-to-vertex shortest path algorithms. With regard to Single Source Shortest Path Problem (SSSP), preprocessing network graphs to facilitate traditional SSSP queries yields a huge performance benefit [14]. Delling and Werneck [15] proposed a customization of the graph that does not only provide much faster query processing but also tries to speed up the graph transformation. In [16], Abraham et al. studied several preprocessing techniques and a measure based on the underlying routing network that quantifies the achievable performance improvement by such a preprocessing.

A combination of both path pre-computation and graph transformation is reach-based routing [17]. Gutman [17] shares with our idea similar considerations about the projection of a network into $\mathbb{R}^2$ and also assumes that there exists a lower bounding metric. With the lower bounding metric in mind, Gutman elaborates an effective pruning method for enhancing the common shortest-path algorithms. In its preparation phase, the reach is computed for all vertices. For that, we take all the shortest paths that pass through a vertex $v$: We calculate the minimum value of the distance between its start vertex and $v$, respectively, the distance between $v$ and its end vertex. Then, the *reach* of $v$ consists of the maximum of all these values. In the evaluation phase of an SSSP algorithm such as Dijkstra or A* [18], discovered vertices are pruned (instead of being put into the candidate list) if their reach is lower than the lower bounding metric.

A more elaborated pruning technique consists of contraction hierarchies [19], which recursively contract vertices to form layers of routing networks of smaller sizes. This technique was refined to so-called customizable contraction hierarchies [20] supporting dynamic changes of the edge costs. It has been recently investigated and improved by Gottesbüren et al. [21] and subsequently by Strasser et al. [22].

Another line of research is hierarchical hub labels [23], in which distances can be inferred by selecting designated hub vertices storing the forward distance from a source and the backward distance to a target. This approach is similar to the techniques presented in this article. The difference is that we have fixed these designated vertices, while the selection of hub vertices is flexible but should cover the whole graph (meaning that for each pair of vertices $(s, t)$, there is a hub vertex on the shortest path from $s$ to $t$). Due to that fact, the drawbacks of hub labels are slow pre-computation times and large index sizes (see [24] for some theoretical bounds). A practical improvement regarding space has been proposed by Delling et al. [25]. Recently, Funke [26] proposed a combination of hub labels with the aforementioned contraction hierarchies. For both techniques, Rupp and Funke [27] proved a lower bound of $\Omega(\sqrt{|V|})$ on the number of average vertex visits for answering the SSSP for both preprocessing techniques.

We note that this list of approaches is far from exhaustive. For instance, a survey on recent preprocessing strategies for the shortest path queries on routing networks is given by Bast et al. [28]. While all these approaches propose indexing data structures for the shortest path queries, the focus of this article is the computation of the ranking of the target vertices, which can be done under certain circumstances without calculating the shortest paths to all targets. In that sense, our approach can be understood as an orthogonal augmentation of approaches solving shortest paths. For instance, preprocessing the routing network with techniques such as contraction hierarchies can help speed up our precomputation steps. To keep the analysis in this article simple, we stick to the well-understood shortest path algorithms of Dijkstra and A* without applying any pre-computation on the routing network.

Another relaxation of the problem of computing exacting shortest distances is distance approximation. Here, Thorup and Zwick [29] proposed a distance approximation function that overestimates the actual distance by at most a fixed factor that can be adjusted in the pre-computation phase. The most recent improvements of this approximation are thanks to Charalampopoulos et al. [30] and Long and Pettie [31].

In the area of skyline computation, Deng et al. [32] also use the Euclidean distance for a lower bound on the actual routing network distance. They restrict the problem of skyline computation to the minimization of distances for multiple spatial objects. By doing so, they can not only share the processing of routing network evaluation on multiple sources but also have much more opportunities for evolving pruning techniques as the skyline attributes are fixed to the distance domain; i.e., their skyline must not contain any other attributes (e.g., textual). Their main idea takes also a lower bounding metric (Euclidean distance) into account for guessing the nearest neighbors on routing networks. See also Section 5.2 in [33] for other multi-criteria approaches.

### 2.1. Our Contribution

In this article, we study the following indexing problem: Given

- A weighted directed graph $(V, A, c)$ with vertices $V$, arcs $A \subset V \times V$, and a cost function $c : A \to \mathbb{R}$, and
- A set of target vertices $\mathcal{T} \subset V$,

find an index of size $\mathcal{O}(|\mathcal{T}|^2)$ that can accelerate the computation of the following query.

DISTANCE RANKING PROBLEM

**Input:** Query vertex $s \in V$;

**Output:** List $L$ of all target vertices sorted with respect to the shortest distances from $s$, i.e.,

- $|L| = |\mathcal{T}|$,
- $L[i] \in \mathcal{T}$, and
- $\rho(s, L[i]) < \rho(s, L[i+1])$ for all $i \in [1..|\mathcal{T}| - 1]$,

where $\rho$ is the quasimetric induced by the cost function $c$.

We define the weighted directed graphs and quasimetrics in Definitions 1 and 2, respectively. We further reformulate this problem in Problem 1 and also give a variation for time-dependent costs in Problem 3. Although the explicit formulation of this problem seems to be new, it has applications as in the aforementioned use cases such as the skyline computation involving the ranking based on distances. Any SSSP algorithm such as those addressed in the related work can solve this problem by computing the exact values of $\rho(s, u)$ for each $u \in \mathcal{T}$. Here, we pose the question of whether we can accelerate the computation: instead of computing the distances to *all* target points, we propose an indexing data structure of size $\mathcal{O}(|\mathcal{T}|^2)$ that helps us to complete the list $L$ by having computed only a few distances $\rho(s, u)$, preferably for the target points with the smallest such values. Our solution works independently from the actual used SSSP solution, which we demonstrate by running an augmented A* and a solution augmenting Dijkstra's algorithm. Since our approach is orthogonal to other preprocessing techniques highlighted in the related work, it is possible to apply our solution on graphs preprocessed with sophisticated techniques such as contraction hierarchies or hub labels. Doing so should speed up our preprocessing step and also the query algorithm (since we need at least one distance value if the query vertex is not among the target vertices). However, to keep the comparison between plain solutions and solutions augmented with our indexing data structure simple, we refrain from using such kinds of techniques, from which we expect to receive relatively equal gains in speed for both the plain and the augmented solution. In addition, note that such techniques need additional preprocessing time, and the space usage can go beyond the $\mathcal{O}(|\mathcal{T}|^2)$ limit.

### 2.2. Structure of the Paper

In the study of the aforementioned fields such as skyline computation and recommender systems, it is often sufficient to obtain a ranking of the target vertices instead of the actual distances. Therefore, we pose the problem to rank a given static set of target vertices by the distances from a query vertex. Our posed problem refrains from taking actual distance computation such as the common SSSP into account. So, is it possible to gain performance by diminishing the expectancy in detailed computation results while supporting pre-computation on the target vertices? We propose in this article a technique based on pre-computation that answers the query without any further computation on the network under certain circumstances. For that purpose, we analyze basic properties of quasimetric spaces in Section 3. In Section 4, we transfer some of these results to quasimetric networks. Further in this section, we introduce the first variant of our estimating algorithm working on a static quasimetric network. Time-dependent quasimetrics along with a time-dependent quasimetric network are introduced in Section 5. In the end of this section, we present derived practices that are applicable to time-dynamic networks. In Section 6, we put the results of Section 4 into practice—an implementation of the algorithm on static networks is described in Section 6.3 and evaluated in Section 6.4. Finally, we summarize our results and draw a conclusion in Section 8.

## 3. Properties of Quasimetric Spaces

Modeling a space with a metric is often too restrictive. There are many problems that can only be expressed with a function that lacks some characteristic of a metric—e.g., reflexivity, non-negativity, symmetry, or the triangle inequality [34]. Inverting a path of an undirected graph results in a path of the same length. However, on directed graphs, this is usually not the case. Hence, the inverse of a shortest path might not be a shortest path. Therefore, we drop the necessity of symmetry and yield a quasimetric:

Let $V$ be a set, which will later serve as the set of vertices of our routing network.

**Definition 1** (Quasimetric). *A mapping $\rho : V \times V \to \mathbb{R}$ is called quasimetric if it satisfies*

$$\rho(u,v) \geq 0 \qquad \text{(non-negativity)},$$
$$\rho(u,v) = 0 \text{ if and only if } u = v \qquad \text{(positive definiteness), and}$$
$$\rho(u,w) \leq \rho(u,v) + \rho(v,w) \qquad \text{(triangle inequality)}.$$

*Furthermore, we call $(V,\rho)$ a quasimetric space. Obviously, whenever $\rho$ is symmetric, we yield a metric space $(V,\rho)$ with $\rho$ as the metric.*

For deeper insight, Mennucci [35] treats quasimetric spaces in detail. The further examinations follow Hetland [36] and Samet [37] by generalizing results for the indexing of metric spaces toward the quasimetric case.

We use the following lemmata to find lower and upper bounds for the quasimetric induced by the cost function of a weighted graph. These bounds are defined by the distance between the query vertex $s \in V$ and a pivot $p \in V$. Therefore, they provide an estimation of $\rho$ for arbitrary vertices $u, v \in V$. Let us keep this notation in mind when we dive into the following inequalities:

**Lemma 1.** *For $p, s, v \in V$, we have the basic inequality*

$$\max\{\rho(s,p) - \rho(v,p), \rho(p,v) - \rho(p,s)\} \leq \rho(s,v).$$

**Proof.** With triangle inequality, we have $\rho(p,v) \leq \rho(p,s) + \rho(q,v)$ and $\rho(s,p) \leq \rho(s,v) + \rho(v,p)$. Transforming both inequalities yields $\rho(p,v) - \rho(p,s) \leq \rho(s,v)$ and $\rho(s,p) - \rho(v,p) \leq \rho(s,v)$, respectively. Figure 1 gives an illustration for these inequalities in the Euclidean space. □

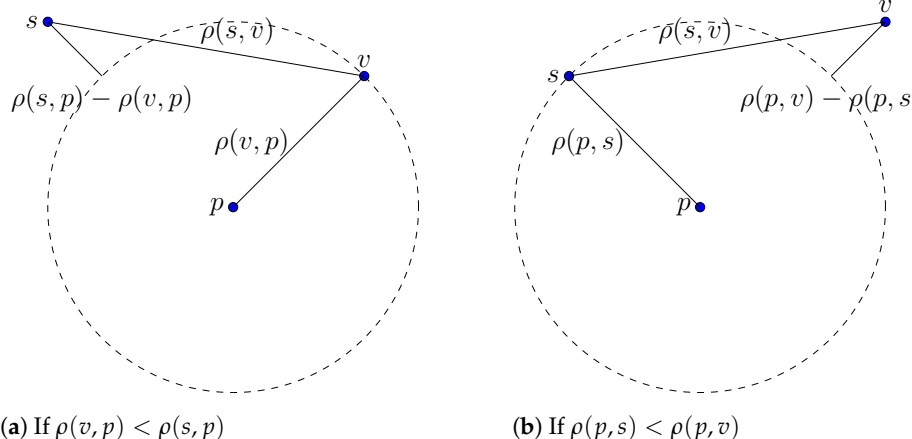

(a) If $\rho(v,p) < \rho(s,p)$        (b) If $\rho(p,s) < \rho(p,v)$

**Figure 1.** Geometrical illustration of the inequality of Lemma 1 for simplicity on the standard Euclidean space.

**Remark 1.** *If $\rho$ is a metric in particular, we get the simplified inequality*

$$|\rho(s,p) - \rho(v,p)| \leq \rho(s,v).$$

**Lemma 2.** *Let $p_1, p_2, s, v \in V$ with $\rho(p_1, v) \leq \rho(p_2, v)$. Then, we have*

$$\rho(p_1, s) - \rho(p_2, s) \leq \rho(s, v) + \rho(v, s).$$

**Proof.** By applying the triangle inequality, we obtain $\rho(p_1, s) \leq \rho(p_1, v) + \rho(v, s)$, and with our assumption, we obtain $\rho(p_1, s) \leq \rho(p_2, v) + \rho(v, s) \leq \rho(p_2, s) + \rho(s, v) + \rho(v, s)$. Rearranging the terms yields the claim. □

**Remark 2.** *We can simplify the result of Lemma 2 for a metric $\rho$ to*

$$\rho(p_1, s) - \rho(p_2, s) \leq 2\rho(v, s).$$

*This inequality is also treated in Lemma 4.4 of [37].*

**Lemma 3.** *Let $p_3, p_4, s, v \in V$ with $\rho(v, p_3) \leq \rho(v, p_4)$. Then, we have*

$$\rho(s, p_3) - \rho(s, p_4) \leq \rho(s, v) + \rho(v, s).$$

**Proof.** Analogously to the proof of Lemma 2. □

**Corollary 1.** *Let $s, v \in V$, $P_v \subseteq \{(p_1, p_2) \in V \times V : \rho(p_1, v) \leq \rho(p_2, v)\}$ and $Q_v \subseteq \{(p_3, p_4) \in V \times V : \rho(v, p_3) \leq \rho(v, p_4)\}$ be sets of pairs. Then, we have*

$$\max_{(p_1, p_2) \in P_v, (p_3, p_4) \in Q_v} \{\rho(p_1, s) - \rho(p_2, s), \rho(s, p_3) - \rho(s, p_4)\} \leq \rho(s, v) + \rho(v, s).$$

**Proposition 1.** *Given two mappings $\hat{\rho}, \check{\rho} : V \times V \to \mathbb{R}$ with*

$$\check{\rho}(u, v) \leq \rho(u, v) \leq \hat{\rho}(u, v) \text{ for all } u, v \in V. \tag{1}$$

*Then, $\rho(s, p_1) < \rho(s, p_2)$ is fulfilled for any $s \in V$ if $\hat{\rho}(s, p_1) < \check{\rho}(s, p_2)$ is satisfied.*

**Proof.** Trivial as $\rho(s, p_1) \leq \hat{\rho}(s, p_1) < \check{\rho}(s, p_2) \leq \rho(s, p_2)$. □

**Example 1.** *For an arbitrary vertex $p \in V$, let*

$$\hat{\rho}(s, v) := \rho(s, p) + \rho(p, v) \text{ and} \tag{2}$$

$$\check{\rho}(s, v) := \max\{\rho(s, p) - \rho(v, p), \rho(p, v) - \rho(p, s)\}. \tag{3}$$

*On the one hand, we observe that $\hat{\rho}$ is a mapping that fulfills the upper bound condition explained in Proposition 1 as $\rho$ holds the triangle inequality. On the other hand, having Lemma 1 in mind, we see that $\check{\rho}$ satisfies the lower bound condition of the proposition.*

**Lemma 4** (δ-Criterion). *Let $\hat{\rho}$ and $\check{\rho}$ be defined as in Example 1. Let us now choose for fixed $p, s \in V$ some $v, w \in V$. Then, for $\delta := \rho(p, s) + \rho(s, p)$, we have*

$$\check{\rho}(s, v) > \hat{\rho}(s, w) \Leftrightarrow \rho(p, v) > \delta + \rho(p, w).$$

**Proof.** If we use the bounds defined as above, we have

$$\check{\rho}(s, v) > \hat{\rho}(s, w) \Leftrightarrow \rho(s, p) - \rho(v, p) > \rho(s, p) + \rho(p, w) \tag{4}$$

$$\vee \rho(p, v) - \rho(p, s) > \rho(s, p) + \rho(p, w). \tag{5}$$

The first inequality on the right side of (4) can be dropped, as $0 > \rho(p, w) + \rho(v, p)$ can never be fulfilled. The latter (5) can be written with as $\rho(p, v) > \delta + \rho(p, w)$. □

The applicability of the $\delta$-criterion for determining the order of two target vertices depends on a good choice of the pivot $p$ close to the start vertex $s$, as can be seen in Figure 2.

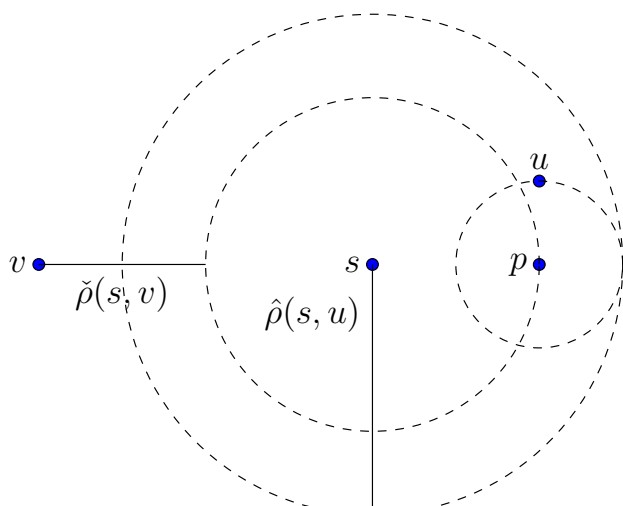

**Figure 2.** Unsuitable condition for the $\delta$-criterion. The success of the $\delta$-criterion basically depends on the distance between query vertex $s$ and pivot $p$. Here, although $u$ is closer to $s$ than $v$, the $\delta$-criteria is in this situation indecisive.

**Example 2.** *We call the mapping $d(u, v) := \rho(u, v) + \rho(v, u)$ the* round trip *induced by $\rho$. It is obvious that $d$ is a metric and thus induces a metric space $(V, d)$. With Corollary 1 (and its notation), we can choose*

$$\check{d}(s, v) := \max_{(p_1, p_2) \in P_v, (p_3, p_4) \in Q_v} \{\rho(p_1, s) - \rho(p_2, s), \rho(s, p_3) - \rho(s, p_4)\} \tag{6}$$

*as the lower bound in Proposition 1. An alternate lower bound for common metrics could be deduced by using Remark 2. It goes without saying that two lower bounds $\check{d}_1, \check{d}_2$ with $\check{d}_1, \check{d}_2 \leq d$ could be combined to a mapping*

$$\check{d}_{12}(u, v) := \max\{\check{d}_1(u, v), \check{d}_2(u, v)\}$$

*that also fulfills the condition of a lower bound. The upper bound can be kept by the same strategy as in Example 1 by means of abusing the triangle inequality.*

## 4. Routing Networks

The following definitions follow the conception of Bondy and Murty [38]. Here, we use the term "simple path" instead of just "path" as it is nowadays sometimes confused with a walk.

**Definition 2** (Directed Weighted Graphs). *A* directed weighted (simple) graph *$G := (V, A, c)$ is a triple consisting of a vertex set $V$, an arc set $A \subseteq V \times V$, and a cost function $c : A \to \mathbb{R}$. For actual computation, we consider a finite graph. Let us denote for two vertices $v_1, v_2 \in V$ with $(v_1, v_2) \in A$ the arc that connects $v_1$ to $v_2$. To avoid an excessive usage of brackets, we simplify the expression $c((u, v))$ to $c(u, v)$ for an arc $(u, v) \in A$. A* walk *$P := (v_1, \ldots, v_n)$ is a consecutive succession of vertices for an arbitrary $n \in \mathbb{N}$ such that there exists an arc $(v_i, v_{i+1}) \in A$ for each $i \in \{1, \ldots, n-1\}$. If the vertices $v_i$ and $v_j$ of a walk $P := (v_1, \ldots, v_n)$ are pairwise different for all $i \neq j$, we call $P$ a* simple path. *We say the walk $P$ consisting of $n \in \mathbb{N}$ vertices "follows from $u \in V$ to $w \in V$" when $v_1 = u$ and $v_n = w$. We call $G$* connected *if there exists a walk from $u$ to $v$ for all $u, v \in V$. Moreover, we define the length of the walk $P = (v_1, \ldots, v_n)$ by $\ell(P) := \sum_{i=1}^{n-1} c(v_i, v_{i+1})$.*

**Definition 3** (Quasimetric Networks). *If A and c of a connected graph $G := (V, A, c)$ support the conditions*

$$(v, v) \in A \text{ for each } v \in V \qquad \text{(reflexivity)},$$
$$c(a) \geq 0 \text{ for each } a \in A \qquad \text{(non-negativity), and}$$
$$c(u, v) = 0 \text{ if and only if } u = v \qquad \text{(definiteness)},$$

*we call G a quasimetric network.*

**Lemma 5.** *Let $(V, A, c)$ be a quasimetric network. By setting*

$$\rho(u, v) := \inf\{\ell(P) : P \text{ simple path from } u \text{ to } v\}$$

*we obtain a quasimetric $\rho$ and call $\rho$ the quasimetric induced by c.*

**Proof.** As a result of $c(a) \geq 0$ for all $a \in A$, a shortest walk is always a simple path: Every walk that contains a circle is not a simple path. It can be made simple by removing all circles. However, this will also shorten the length of the walk. Hence, the definition of $\rho$ remains the same when taking the infimum of the length of all walks. We call a walk that meets this condition a *shortest path*. Hence, we yield:

- From $l \geq 0$, we get the non-negativity of $\rho$.
- As $\ell(P) > 0$ for all walks from $u$ to $v \in V, u \neq v$, we yield by definition $\rho(u, v) > 0$ for all $u, v \in V, u \neq v$. As $c(v, v) = 0$ for each $v \in V$, we can conclude that $\ell((v, v)) = 0$ for each $v \in V$. Thus, we yield the positive definiteness of $\rho$.
- If we define the concatenation of walks by

$$(v_1, \ldots, v_n) \circ (w_1, \ldots, w_n) := \begin{cases} (v_1, \ldots, v_n, w_2, \ldots, w_n) & \text{if } v_n = w_1, \\ \text{undefined} & \text{otherwise,} \end{cases}$$

triangle inequality is simple to show: For arbitrary $u, v, w \in V$, let $P_{uv}$ be a simple path from $u$ to $v$, and $P_{vw}$ be a simple path from $v$ to $w$. Then, we can generate a walk $P_{uw} := P_{uv} \circ P_{vw}$ by combining both paths, so we have $\ell(P_{uw}) = \ell(P_{vw}) + \ell(P_{uv})$. Applying the infimum over all walks from $u$ to $w$ yields the triangle inequality. □

Until the end of this section, let us use the symbol $\rho$ for the quasimetric implicitly induced by the cost function of a given quasimetric network.

**Definition 4** (Lower bounding metric). *Let $(V, A, c)$ be a quasimetric network. We call a metric $d : V \times V \to \mathbb{R}$ which satisfies the condition*

$$d(u, v) \leq \rho(u, v) \text{ for all vertices } u, v \in V \qquad \text{(beeline property)},$$

*a lower bounding metric of $\rho$.*

**Remark 3** (Preparation). *We assumed that $\mathcal{T}$ is relatively tiny compared to $V$, such that the pre-computation task stated beyond can be processed in a feasible amount of time and space. Let $\iota : V \to \mathbb{N}$ be an enumeration of all vertices. We cache all values of the mapping $\iota(\mathcal{T}) \times \iota(\mathcal{T}) \to \mathbb{R}, (i, j) \mapsto \rho(\iota^{-1}(i), \iota^{-1}(j))$ in a matrix which we name M. M is asymmetric if $\rho \mid_{\mathcal{T} \times \mathcal{T}}$ is not symmetric, i.e., is not a metric. For the next steps, we will not explicitly refer to this matrix. Whenever a value of type $\rho(v, w)$ with $v, w \in \mathcal{T}$ is requested, it is now clear that we can look up this value in M with $\mathcal{O}(1)$ time.*

For our problem, the lower and upper bounds acquired in Section 3 are especially useful if we find a vertex $p \in V$ for which we know the distance between $p$ and the query vertex $s$ and the distances between $p$ and every target.

**Definition 5** (Pivot vertex). *A vertex is called a* pivot vertex *if its distance to the query vertex s as well as its distance to every target vertex is known. Let us denote with $P \subset V$ the set of pivot vertices.*

Due to Remark 3, we already have $\mathcal{T} \subset P$ at the beginning of the algorithm. We now state the problem we want to tackle in this article:

**Problem 1** (Shortest Path Ranking Problem). *Let $G = (V, A, c)$ be a quasimetric network, $V$ be finite, and $\mathcal{T} \subset V$ with $|\mathcal{T}| \ll |V|$ a small subset of $V$. Given a query vertex $s \in V$, we ask for the relation $\leq_s$ given by the equivalence*

$$u \leq_s v \Leftrightarrow \rho(s, u) \leq \rho(s, v) \; \forall u, v \in \mathcal{T}.$$

**Remark 4.** *It is obvious that $\leq_s$ is a weak ordering on $\mathcal{T}$, so especially transitivity holds. A trivial solution would be to just compute the shortest paths from s to all targets so that $\rho(s, p)$ is known for all $p \in \mathcal{T}$. Let us imagine that we could obtain some bounds that approximate the values of $\rho$, i.e., two functions whose values are fast to compute. These values would give us an interval in which we can expect the actual value of $\rho$. Then, we can calculate the relation $\leq_s$ on behalf of this approximation. Therefore, if we consider that calculation of the net distances is expensive, whereas the Euclidean distances between vertices in a routing network are trivial to get, we could use the lower bounding metric d (e.g., the Euclidean norm $d(u, v) = \|u - v\|_2$ if we can identify vertices of V with some points of a real vector space) for our algorithm.*

**Solution 1.** *Let $U$ be the set of all targets for which we do not know the relationship $\leq_s$ to each other element of $\mathcal{T}$. We generate U by defining a strict weak ordering $R_s$ and an equivalence relation $E_s$ with our incomplete knowledge about $\leq_s$ such that $p \in U :\Leftrightarrow pR_s \cup R_s p \cup E_s p \neq \mathcal{T} \setminus \{p\}$. If $U = \emptyset$, we are already done. This is immediately the case if we luckily get a query vertex $s \in \mathcal{T}$, because then, all the necessary routing data are already present in the saved matrix M. Otherwise, our employed SSSP algorithm has to find out at least one shortest path from s to any target in $\mathcal{T}$. If we have a lower bounding metric and an SSSP algorithm that employs this metric for heuristic search, we create a priority heap and order the set of targets by the lower bounding metric from the query vertex. We modify the shortest path finder such that it takes the closest element off the heap and applies its heuristic according to this target. Hence, the search will firstly focus on finding shortest paths from s to $p := \mathrm{argmin}_{v \in \mathcal{T}} d(s, v)$. To use our bounds, we have to obtain the values of $\rho(s, p)$ and $\rho(p, s)$. Therefore, we start with a shortest path search from s to p and vice versa (hence, p becomes an element of P). With respect to s and p, we now define the distances $\hat{\rho}, \check{\rho}$ such as in Example 1. We check for all pairwise different tuples $(u, v) \in U \times U$, if we now perceive a relationship between u and v with respect to $\leq_s$: If $\hat{\rho}(s, u) < \check{\rho}(s, v)$ holds, we have $u <_s v$ (i.e., $u \leq_s v$ and $v \not\leq_s u$). Hence, we delete each element u from U for which we know the relationship $\leq_s$ between u and all other targets. This is the case when $|uR_s \cup R_s u \cup E_s u| = |\mathcal{T}| - 1$. If there are still some elements remaining in U, we continue our started SSSP algorithm and try to find the next closest target contained by U. The algorithm prematurely terminates if it has gathered enough information about the missing relationships. Thus, a full graph search is often not required. For example, when we have reached a target $p' \in \mathcal{T} \setminus \{p\}$, we already know the lengths of certain walks from s to all other targets thanks to the matrix M. If we have also found $\rho(p', s)$ by coincidence, we can generate new bounds $\hat{\rho}_{p'}$ and $\check{\rho}_{p'}$ according to $p'$ that hold the bounding condition (1) again. Therefore, we lower the upper bound*

$$\hat{\rho}'(s, v) := \min\left\{ \hat{\rho}(s, v), \hat{\rho}_{p'}(s, v) \right\} \geq \rho(s, v) \tag{7}$$

*and raise the lower bound*

$$\check{\rho}'(s, v) := \max\left\{ \check{\rho}(s, v), \check{\rho}_{p'}(s, v) \right\} \leq \rho(s, v) \tag{8}$$

*for all $v \in \mathcal{T} \setminus \{p, p'\}$. Next, we check if the better bounds solve any yet missing relationship, since we are looking forward to removing all vertices from U. Finally, the algorithm stops when $U = \emptyset$. In another perspective, $R_s$ and $E_s$ can be represented as a set of Hasse diagrams that we incrementally adhere to a single Hasse diagram representing the weak ordering $\leq_s$.*

Algorithm 1 shows an exemplary implementation with the bounds considered in Example 1 and Lemma 6. The algorithm represents both relations it computes as graphs. The idea of Algorithm 1 is to maintain a dynamic set $U$ (Line 4) of all target vertices whose order has not yet been determined. Whenever the applied SSSP algorithm (Line 5) finds a shortest path from $s$ to one of the vertices of $U$, the algorithm improves the upper and lower bounds $\hat{\rho}$ and $\check{\rho}$, and it tries to determine the order of the remaining elements of $\mathcal{U}$ by calling COMPUTERELATIONSHIPS of Algorithm 2 in Line 12. In Algorithm 2, we first determine the order of those target vertices whose distances from the start vertex have become known (Line 3) or whose order we can determine with the improved upper and lower bounds (Line 8). Finally, we remove those target vertices from $U$ (Line 13).

---

**Algorithm 1** An implementation on quasimetric networks

**Require:** $G$ : Quasimetric Network, $\mathcal{T} \subset V$

**Ensure:** Return value is the ordering relation $(<_s, =_s)$

1: **function** QUERY($s$ : Vertex)
2:     $R_s \leftarrow Digraph(\mathcal{T}, \emptyset)$                              ▷ relation $<_s$ as a directed graph
3:     $E_s \leftarrow Graph(\mathcal{T}, \emptyset)$                                        ▷ relation $=_s$
4:     $U \leftarrow \mathcal{T}$                    ▷ set of target vertices whose order is yet unknown
5:     $S \leftarrow$ Instance of an SSSP solver on $G$
6:     **while** $U \neq \emptyset$ **do**          ▷ if $U = \emptyset$, we know the order of all target vertices
7:        $p \leftarrow S.findOneOf(U)$                  ▷ get the next pivot vertex $p$
8:        **if** $\hat{\rho}$ and $\check{\rho}$ are undefined **then**
9:           $\hat{\rho} \leftarrow (2)$ and $\check{\rho} \leftarrow (3)$
10:       **else**
11:          $\hat{\rho} \leftarrow (10)$ and $\check{\rho} \leftarrow (11)$             ▷ apply $\gamma$-criterion
12:        $U \leftarrow$ COMPUTERELATIONSHIPS($R_s, E_s, U, s$)
13:     **return** $(R_s, E_s)$

---

**Algorithm 2** Complete the ranking that is represented by both graphs

1: **function** COMPUTERELATIONSHIPS($R_s, E_s, U, s$)
2:     **for all** $(v, w) \in U \times U$ pairwise different **do**
3:        **if** $\rho(s, v)$ and $\rho(s, w)$ are known **then**
4:           **if** $\rho(s, v) = \rho(s, w)$ **then**
5:              $E_s.addEdge(w, v)$
6:           **else if** $\rho(s, v) > \rho(s, w)$ **then**
7:              $R_s.addArc(w, v)$
8:        **else if** $\check{\rho}(s, v) > \hat{\rho}(s, w)$ **then**
9:           $R_s.addArc(w, v)$
10:     **for all** $v \in U$ **do**
11:        $S \leftarrow R_s.neighborsOf(v) \cup E_s.neighborsOf(v)$
12:        **if** $|S| = |\mathcal{T}| - 1$ **then**
13:           $U \leftarrow U \setminus \{v\}$
14:     **return** $U$

---

**Example 3.** *Let us again consider Example 1 with $\rho(s, v) \neq \rho(s, w)$ for all $u, w \in \mathcal{T}$ pairwise different. Moreover, let $p \in P$ be given. Furthermore, $p$ shall induce the bounds $\hat{\rho}$ and $\check{\rho}$. If $\delta$ is sufficiently small enough, such that $\delta < \min_{v,w \in \mathcal{T}: v \neq p \neq w} |\rho(p, v) - \rho(p, w)|$, then for all $v, w \in \mathcal{T}$, the condition $\check{\rho}(s, v) > \hat{\rho}(s, w)$ holds if and only if $\rho(p, v) > \delta + \rho(p, w)$. However, if*

$\delta \geq \max_{v,w \in \mathcal{T}} |\rho(p,v) - \rho(p,w)|$, *we cannot determine a single relationship between two targets in the beginning.*

**Remark 5** (Misleading Heuristic). *We assumed in Solution 1 that d will show us the target which is actually closest to s. However, this might not be the case in general. In the worst case, we might collect shortest path distances to all other targets before actually finding the closest target with respect to d. We can avoid this by discarding p when we encounter another target p' with probably $\rho(s,p') \leq \rho(s,p)$ and hopefully small $\rho(p',s)$ while computing the shortest paths from s to p.*

**Remark 6** (Consecutive Queries). *The algorithm can be extended with an online component to cache query results for which the algorithm could not guess any order; i.e., all distances had to be computed. More concretely, this means that at the end of the query process, for a query vertex s, the database has computed the distances $\rho(s,p)$ and $\rho(p,s)$ for all $p \in \mathcal{T}$. We save the vertex s along with its distances in a set $Q \subset V \times \mathbb{R}^{2|\mathcal{T}|}$. This set is now used along with the matrix M for building the bounds: therefore, for every new query $s' \in V$ we check whether s' is closer to a vertex $s \in \pi_V Q \subset V$ than any target of $\mathcal{T}$ with respect to d. If this is the case, we define our bounds $\check{\rho}$ and $\hat{\rho}$ according to s, which is now an element of P.*

**Remark 7** (ALT algorithm). *The algorithm using A\* search, landmarks, and the triangle inequality (ALT) by Goldberg and Harrelson [39] is an SSSP algorithm, which regards similar aspects: Their routing algorithm involves the pre-calculation of distances between a given landmark set $L \subseteq V$ and all vertices of V. The set of pre-calculated values is*

$$\{(l,v,\rho(l,v)) : l \in L, v \in V\} \cup \{(v,l,\rho(v,l)) : l \in L, v \in V\}.$$

*Hence, $L \subseteq P$. For some fixed $l \in L$ and arbitrary $p,s \in V$, we split up the definition of $\check{\rho}$ in Example 1 by $\check{\rho}(s,p) = \max\{h_1(s),h_2(s)\}$ where $h_1,h_2 : V \to \mathbb{R}$ with $h_1(s) := \rho(s,l) - \rho(p,l)$ and $h_2(s) := \rho(l,p) - \rho(l,s)$. $h_1,h_2$ are admissible/feasible heuristic functions, i.e., $c(v,w) + h_j(w) - h_j(v) \geq 0$ holds for all $j = 1,2$ and for all $(v,w) \in A$ because $c(v,w) + h_1(w) - h_1(v) = \rho(v,w) + \rho(w,l) - \rho(v,l) \geq 0$ and $c(v,w) + h_2(w) - h_2(v) = \rho(v,w) - \rho(l,w) + \rho(l,v) \geq 0$. Lemma 2.1 of Goldberg and Harrelson [39] shows that the maximum of two feasible functions is again feasible. Hence, $\check{\rho}$ with a fixed $l \in L$ can be used as a heuristic function for processing A\*. For each $p \in L \cap \mathcal{T} \neq \emptyset$, we do not only know the distances $\rho(p,p') \forall p' \in \mathcal{T}$ but also $\rho(v,p) \forall v \in V$. In particular, for $\mathcal{T} \subseteq L$, our problem is already trivially solved.*

*Possible Extensions*

**Remark 8** (Non-connected graphs). *Generated graphs from real data are in general not connected. Luckily, we can tweak our algorithm with a small modification in order to cope with inaccessible parts of the (not necessarily connected) routing network. For example, let us take a pivot $p \in P$ and a vertex $v \in \mathcal{T}$ that is inaccessible by p, i.e., $\rho(p,v) = \infty$. Furthermore, let us consider that we could find a shortest path from p to the start s. Then, by triangle inequality $\rho(p,v) \leq \rho(p,s) + \rho(s,v)$, we have $\rho(s,v) = \infty$. Hence, we already know that v is inaccessible. Let us consider a graph in which for each arc $(u,v) \in A$, the converse $(v,u) \in A$ exists. Then, by the result above, we do not have to exhaustively traverse the graph to determine the inaccessibility of a certain target (if we have accessed at least one pivot vertex).*

**Remark 9** (Validity regions). *It is common that multiple queries are issued in the vicinity of the last query vertex. In particular, moving objects that frequently want to fetch updates for their last query results tend to continuously poll the database for changes. To minimize data traffic and computation on the server side, the database server could deliver a validity region along with the query result. A* validity region *is a set of vertices for which we get the same query result as for the latest query vertex [40]. With this information, a client can keep the last query information as long as the query vertex does not leave the validity region. A priori, if we find a sufficiently small $\delta$*

*with regard to a $p \in \mathcal{T}$ as in Example 3, we are already done with the computation and could verify identical results for all query vertices that are contained in a region of*

$$\left\{ s \in V : \rho(s,p) + \rho(p,s) < \min_{v,w \in \mathcal{T}: v \neq p \neq w} |\rho(p,v) - \rho(p,w)| \right\}. \tag{9}$$

*Let us assume the case that we could not find such a $\delta$. Then, we could not compute the complete relation and thus could not solve the problem beforehand. That means that we have encountered at least one other target before our algorithm quit successfully.*

Now, let $p, p' \in \mathcal{T}$ be two arbitrary chosen targets that we have processed and found while computing the shortest paths. Then, the following lemma gives us a new criterion:

**Lemma 6** ($\gamma$-Criterion). *Let $s \in V$ and $p, p' \in \mathcal{T}$ be given. Furthermore, let $\hat{\rho}$ and $\check{\rho}$ be defined as (2) and (3), respectively, according to $p$. As in Solution 1, we improve the lower and upper bounds of the quasimetric $\rho$ by*

$$\hat{\rho}'(s,v) := \min\{\hat{\rho}(s,v), \rho(s,p') + \rho(p',v)\} \geq \rho(s,v) \text{ and} \tag{10}$$
$$\check{\rho}'(s,v) := \max\{\check{\rho}(s,v), \rho(s,p') - \rho(v,p'), \rho(p',v) - \rho(p',s)\} \leq \rho(s,v). \tag{11}$$

*Now, let us choose some $v, w \in \mathcal{T}$. Then, for $\gamma_{p,p'} := \rho(p,s) + \rho(s,p')$, we have*

$$\check{\rho}'(s,v) > \hat{\rho}'(s,w) \Leftarrow \rho(p,v) > \gamma_{p,p'} + \rho(p',w).$$

**Proof.** Analogously as Lemma 4, we want to verify the relation between two arbitrary $v, w \in \mathcal{T}$ by

$$\check{\rho}'(s,v) > \hat{\rho}'(s,w) \Leftrightarrow \rho(p,v) - \rho(p,s) > \rho(s,p) + \rho(p,w) \tag{12}$$
$$\vee \rho(p',v) - \rho(p',s) > \rho(s,p') + \rho(p',w) \tag{13}$$
$$\vee \rho(s,p) - \rho(v,p) > \rho(s,p') + \rho(p',w) \tag{14}$$
$$\vee \rho(s,p') - \rho(v,p') > \rho(s,p) + \rho(p,w) \tag{15}$$
$$\vee \rho(p',v) - \rho(p',s) > \rho(s,p) + \rho(p,w) \tag{16}$$
$$\vee \rho(p,v) - \rho(p,s) > \rho(s,p') + \rho(p',w), \tag{17}$$

where we have already left out two impossible conditions on the right side similarly to in the proof of Example 1. Moreover, we can similarly transform for $\delta_u := \rho(u,s) + \rho(s,u)$ with $u \in V$ the conditions (12) and (13) to (12) $\Leftrightarrow \rho(p,v) > \delta_p + \rho(p,w)$ and (13) $\Leftrightarrow \rho(p',v) > \delta_{p'} + \rho(p',w)$. Both (12) and (13) are already used in Algorithm 1 for determining the ranking. If we let $\Delta_{u,v} := \rho(s,u) - \rho(s,v)$ be a measure of how much more $u$ is further away from $s$ than $v$ when starting in $s$, then we could write (14) $\Leftrightarrow \Delta_{p,p'} > \rho(p',w) + \rho(v,p)$ and (15) $\Leftrightarrow \Delta_{p',p} > \rho(p,w) + \rho(v,p')$. A $\Delta_{p,p'} > 0$ shows that $p'$ can be more easily reached from $s$ than $p$. Therefore, if $w$ is near enough to $p'$ and thus to $s$ while $v$ is near enough to $p$, then we see from (14) that $w$ can be more easily reached from $s$ than $v$. The converse case can be applied with (15). For the last two inequations, let $\gamma_{u,v} := \rho(u,s) + \rho(s,v)$ be the shortest path from $u$ to $v$ while taking a stopover at $s$. Then, we have (16) $\Leftrightarrow \rho(p',v) > \gamma_{p',p} + \rho(p,w)$ and (17) $\Leftrightarrow \rho(p,v) > \gamma_{p,p'} + \rho(p',w)$. We can think of these equations as generalizations of (12) and (13) taking the pairing of $p$ and $p'$ into account (we can deduct the former formulae by setting $p = p'$). $\quad\square$

**Remark 10.** *Let us recall that unless our routing network is symmetric, we need both values $\rho(p,s)$ and $\rho(s,p)$ for which we want to evaluate $\gamma_{p,\cdot}$ or $\delta_\cdot$ for each $p \in \mathcal{T}$. Clearly, a $\Delta_{p',p} \gg 0$, as tiny values of $\delta_p$ and $\gamma_{p,p'}$ are favorable conditions for determining the ordering relation without a*

*complete shortest path computation.* $\gamma_{p,p'}$ *might be a better measure for a validity region than* $\delta_p$ *if* $\rho(s,p) > \rho(s,p')$. *Again,*

$$\gamma_{p,p'} < \min_{v,w \in \mathcal{T}: \, v,w \notin \{p,p'\}} \max\{\rho(p,v) - \rho(p',w), \rho(p',v) - \rho(p,w)\}$$

*is a sufficient condition to generate the complete ranking order while knowing only the distances between s and p, and s and p'. Thus, a validity region could be returned as a subset of the union of* ((9)—$\hat{\rho}, \check{\rho}$ *defined w.r.t.* p), ((9)—$\hat{\rho}, \check{\rho}$ *defined w.r.t.* p') *and*

$$\bigcup_{p,p' \in \mathcal{T}} \Big\{ s \in V : \rho(p,s) + \rho(s,p') <$$

$$\min_{v,w \in \mathcal{T}: \, v,w \notin \{p,p'\}} \max\{\rho(p,v) - \rho(p',w), \rho(p',v) - \rho(p,w)\} \Big\}$$

**Remark 11.** *Taking the importance of co-location into account, Shekhar and Huang [41] propose the clustering of targets with similar keywords that are in the near vicinity. Following this semantic consideration, we can take advantage of this clustering for our approach. Grouping conglomerations of targets to a single target will uncompress the distances between target groups and thus provide a better chance for determining the ranking beforehand.*

**Problem 2** (Round Trip Problem). *Let us assume the same conditions as for Problem 1, but now, we prefer to retrieve the relation* $\leq_q$ *with respect to the round trip metric* $d(u,v) := \rho(u,v) + \rho(v,u)$. *Clearly, we can use the same algorithm by keeping Example 2 in mind.*

## 5. Time-Dependent Routing Networks

Most real-world scenarios have to cope with the additional factor of time for choosing shortest paths. Rush hours, railroad crossings with gates, or tides at coastal areas can completely reshape the routing situation. Daytime and season should not be neglected for consideration when choosing a route. Hence, a model that represents a dynamic network could fit this problem better than keeping analysis on static networks.

Although Time-Dependent Shortest Path (TDSP) problems [42] have been in vogue lately, Dreyfus [43] has already considered solving the shortest path problem on dynamic networks in 1969. More recently, Chabini and Lan [44] adapted the classic A* algorithm to time-dependent networks. Nannicini et al. [45], Delling and Nannicini [46] refined this approach by adding a bidirectional component to time-aware A* routing. Finally, Strasser [47], Strasser and Zeitz [48] presented improvements by using contraction hierarchies. Other time-dependent shortest path algorithms were proposed by George et al. [49]. In the setting that the full information of the edge costs dependent on the time is unknown, Zhao et al. [50] gave an algorithm based on a Markov model based on a past timeline to predict the temporally changing edge costs.

We conclude that efficient Time-Dependent Shortest-Path algorithms already exist. The notion of shortest path has been transferred to the shortest travel time in the field of time-dependent routing networks. Now, we want to see if we can manage to lift our above propagated algorithm to a modified version of our posed problem that is now based on time-dependent networks. Therefore, let us start with the fundamentals once again:

Let $V$ be a set, $I = [t_0, t_\infty) \subset \mathbb{R}$ an interval with $-\infty < t_0 < t_\infty \leq \infty$.

**Definition 6** (Time-Dependent Quasimetrics). *A mapping* $\rho : I \times V \times V \to \mathbb{R}$ *is called a* time-dependent quasimetric *if it satisfies for all* $t \in I$

$$\rho^t(u,v) \geq 0 \qquad \qquad \text{(non-negativity),} \qquad (18)$$

$$\rho^t(u,v) = 0 \text{ if and only if } u = v \qquad \text{(positive definiteness), and} \qquad (19)$$

$$\rho^t(u,w) \leq \rho^t(u,v) + \rho^{t+\rho^t(u,v)}(v,w) \qquad \text{(time-dep. triangle inequality),} \qquad (20)$$

*if additionally $t + \rho^t(u, v) \in I$. For the sake of readability, we write $\rho^t$ instead of $\rho(t, \cdot, \cdot)$. Furthermore, we call $(V, I, \rho)$ a time-dependent quasimetric space.*

Let $I_{v,w}^s := \left\{ t \in I : t - \rho^t(v, s) \in I \ \wedge \ t + \rho^t(s, w) \in I \right\} \subseteq I$ for $v, w, s \in V$.

**Lemma 7.** *For $p, s, v \in V$ we have the basic inequality*

$$\max\left\{ \rho^t(s, p) - \rho^{t+\rho^t(s,v)}(v, p), \rho^{t-\rho^t(p,s)}(p, v) - \rho^{t-\rho^t(p,s)}(p, s) \right\} \leq \rho^t(s, v)$$

*for every $t \in I_{p,v}^s$.*

**Proof.** We follow the proof of Lemma 1: By time-dependent triangle inequality, we have $\rho^t(p, v) \leq \rho^t(p, s) + \rho^{t+\rho^t(p,s)}(s, v)$ and $\rho^t(s, p) \leq \rho^t(s, v) + \rho^{t+\rho^t(s,v)}(v, p)$. As the first inequality is satisfied for all $t \in \left[ t_0, t_\infty - \rho^t(p, s) \right)$, we can use the translation $t \mapsto t - \rho^t(p, s)$ to get $\rho^{t-\rho^t(p,s)}(p, v) - \rho^{t-\rho^t(p,s)}(p, s) \leq \rho^t(s, v)$. □

**Example 4.** *For an arbitrary vertex $p \in V$, set*

$$\hat{\rho}^t(s, v) := \rho^t(s, p) + \rho^{t+\rho^t(q,p)}(p, v) \text{ and} \tag{21}$$

$$\check{\rho}^t(s, v) := \max\left\{ \rho^t(s, p) - \rho^{t+\rho^t(s,v)}(v, p), \rho^{t-\rho^t(p,s)}(p, v) - \rho^{t-\rho^t(p,s)}(p, s) \right\} \tag{22}$$

*with $\hat{\rho}^t : \left[ t_0, t_\infty - \rho^t(s, p) \right) \times V \times V \to I$ and $\check{\rho}^t : \left[ t_0 + \rho^t(p, s), t_\infty - \rho^t(s, v) \right) \times V \times V \to I$. By Lemma 7, we see that both newly defined functions are feasible bounds of $\rho^t$ that fulfill the bounding condition (1). Now, we want to see whether the $\delta$-criterion somehow also holds in our extended case. For fixed $p, s \in V$, let us choose some $v, w \in V$. Then, we get*

$$\check{\rho}^t(s, v) > \hat{\rho}^t(s, w)$$
$$\Leftrightarrow \rho^t(s, p) - \rho^{t+\rho^t(s,p)}(v, p) > \rho^t(s, p) + \rho^{t+\rho^t(s,p)}(p, w)$$
$$\vee \rho^{t-\rho^t(s,p)}(p, v) - \rho^{t-\rho^t(s,p)}(p, s) > \rho^t(s, p) + \rho^{t+\rho^t(s,p)}(p, w).$$

*We drop the first inequality, because $0 > \rho^{t+\rho^t(s,p)}(v, p) + \rho^{t+\rho^t(s,p)}(p, w)$ can never be fulfilled. If we use the parametrization $t \mapsto \delta_t := \rho^t(p, s) + \rho^{t-\rho^t(p,s)}(s, p)$, we get*

$$\rho^{t-\rho^t(p,s)}(p, o) > \delta_t + \rho^{t+\rho^t(s,p)}(p, w).$$

*Finally, we conclude that for fixed $t \in I_{p,p}^s$, we yield a time-parametrized $\delta_t$-criterion. Thus, again, for sufficiently small $\delta_t$ and a suitable spatial distribution of $\mathcal{T}$, we can solve the time-dependent ranking without solving the TDSP. If $\max_{t \in [t_0+\rho^t(s,p), t_\infty)} \delta_t$ is sufficiently small, we can even solve the problem for all times beforehand.*

**Definition 7** (Time-Dependent Quasimetric Network). *A time-dependent quasimetric network $G$ consists of a tuple $(V, A, I, c)$, where $(V, A)$ is a connected directed graph, $I = [t_0, t_\infty] \subseteq \mathbb{R}$ is an interval, and $c : A \times I \to I$ a cost function for which $G^t := (V, A, c^t)$ is a quasimetric network for all $t \in I$, where $c^t(\cdot) := c(\cdot, t)$. We further write $c_a(\cdot) := c(a, \cdot)$ for any arc $a \in A$. Let $P = (v_1, \ldots, v_n)$ be a walk with $n \in \mathbb{N}$ arbitrary and $v_i, v_j$ be pairwise different for all $i \neq j$. Exactly as Demiryurek et al. [42], we describe the length of $P$ by*

$$\ell^t(P) := \begin{cases} \sum_{i=1}^{n-1} c_{(v_i, v_{i+1})}(t_i) & \text{if } \sum_{i=1}^{n-1} c_{(v_i, v_{i+1})}(t_i) \in I, \\ \infty & \text{otherwise} \end{cases}$$

with $t_1 = t \geq t_0$ and $t_{i+1} = t_i + c_{(v_i,v_{i+1})}(t_i) < t_\infty$ *for each* $i = 1, \ldots, n-1$. *Unfortunately, this definition hides the time dependencies involved for actual computing. For more clarity, we define the* cumulative cost function $c'_a(t) := t + c_a(t)$. *Then, we can rewrite the definition of* $\ell^t$ *by*

$$\ell^t(P) = c_{(v_{n-1},v_n)}\left(c'_{(v_{n-2},v_{n-1})}\left(\ldots c'_{(v_2,v_3)}\left(c'_{(v_1,v_2)}(t)\right)\ldots\right)\right) \text{ if } \ell^t(P) < \infty.$$

**Lemma 8.** *A time-dependent quasimetric network* $(V, A, I, c)$ *induces the time-dependent quasimetric*

$$\rho^t(u, v) := \inf\{\ell^t(P) : P \text{ simple path from } u \text{ to } v\}.$$

*by taking the infimum over the length of all simple paths from* $u \in V$ *to* $v \in V$.

**Proof.** Since $G^t$ is a quasimetric network for all $t \in I$, it is easy to see that $\rho^t$ is a quasimetric for fixed $t \in I$. Therefore, we yield (18) and (19). However, we are not interested in routing with constant $t$. Instead, we can use the notion of $\ell^t$ to obtain (20): Let $t \mapsto P^t_{uv}$ be a parametrization of a shortest path from $u \in V$ to $v \in V$ at time $t \in I$. Then, we have $\ell^t\left(P^t_{uv}\right) = \rho^t(u, v)$. For an additional $w \in V$, we conclude that $\ell^{t+\rho^t(u,v)}\left(P^{t+\rho^t(u,v)}_{vw}\right) = \rho^{t+\rho^t(u,v)}(v, w)$. We can adhere the paths $P^t_{uv}$ and $P^{t+\rho^t(u,v)}_{vw}$ to a walk $P^t_{uvw} = P^t_{uv} \circ P^{t+\rho^t(u,v)}_{vw}$ for which $\ell^t\left(P^t_{uvw}\right) \geq \rho^t(u, w)$ holds and thus $\rho^t(u, w) \leq \rho^t(u, v) + \rho^{t+\rho^t(u,v)}(v, w)$. This finishes the proof that $\rho^t$ is a time-dependent quasimetric. $\square$

Until the end of this article, let us implicitly use the symbol $t \mapsto \rho^t$ for the time-dependent quasimetric induced by the cost function of a given time-dependent quasimetric network.

**Definition 8** (Time-Dependent Lower Bounding Metric). *Let* $(V, A, I, c)$ *be a time-dependent quasimetric network. We call a metric* $d^t : V \times V \to \mathbb{R}$ *that fulfills the condition*

$$d^t(u, v) \leq \rho^t(u, v) \text{ for all vertices } u, v \in V \text{ and for every } t \in I,$$

*the* time-dependent lower bounding metric *of* $\rho^t$. *By definition,* $d^t$ *is a lower bounding metric of* $\rho^t$ *for every fixed* $t \in I$.

**Example 5.** *Let us examine some lower bounds for a time-dependent quasimetric network* $(V, A, I, c)$.
- *If we can pre-compute and store values of routing distances, we can create the quasimetric network* $(V, A, c_{\min})$ *with a new function* $c_{\min} := \inf_{t \in I} c^t$. *Then, the induced quasimetric* $\rho_{\min}$ *is a lower bound of* $\rho^t$ *for all* $t \in I$.
- *If* $c^t(u, v)$ *measures the time it takes for an object* $o$ *to get from* $u \in V$ *to* $w \in V$ *when starting at time* $t$, *then there exists some mapping* $d_2 : V \times V \to \mathbb{R}^n$ *with* $n \in \mathbb{N}$ *so that* $d_2(u, w)$ *measures the distance from* $u$ *to* $w$. *If we know the maximum speed* $v_{\max}(u, w)$ *of* $o$ *while moving from* $u$ *to* $w$, *we can take* $d^t(u, w) := \frac{d_2(u,w)}{v_{\max}(u,w)} \leq c^t(u, w)$ *as a time-independent lower bounding metric.*

**Definition 9** (Discretizable). *Let* $G = (V, A, I, c)$ *be a time-dependent quasimetric network and* $\mathcal{T} \subset V$ *a small set of vertices. For pre-computation and storing of routing evaluation for all times* $t \in I$, *we take additional assumptions:*
- $t \mapsto c_a(t) \in \mathcal{C}$ *for every* $a \in A$ *where* $\mathcal{C} \subset \{f : I \to I\}$ *is a class of functions for which* $\mathcal{C}$ *is closed under composition and the evaluation of a function of* $\mathcal{C}$ *is reasonably fast.*
- *It is possible to divide* $I$ *into* $n \in \mathbb{N}$ *disjoint intervals*

$$[t_1, t_2) \cup [t_2, t_3) \cup \ldots \cup [t_{n-1}, t_n) = I.$$

- *For all* $u, v \in V$ *exist some functions* $f^{(u,v)}_1, \ldots, f^{(u,v)}_n \in \mathcal{C}$ *such that for every* $k \in \{1, \ldots, n-1\}$, *we have* $\rho^t(u, v) = f^{(u,v)}_k(t)$ *for all* $t \in [t_k, t_{k+1})$.

*If these conditions are satisfied, we call the network* discretizable. *Transferring the idea of Remark 3 to a discretizable time-dependent quasimetric network can be done by defining the matrix of distances $M$ with $M : \{1, \dots, n-1\} \to \mathcal{C}^{|\mathcal{T}| \times |\mathcal{T}|}$ by*

$$M(k) := \left( f_k^{(p_i, p_j)} \right)_{i,j \in \iota(\mathcal{T})} \quad \text{with } p_i := \iota^{-1}(i), \, p_j := \iota^{-1}(j) \in \mathcal{T}$$

*for each $k \in \{1, \dots, n-1\}$, where $\iota$ is defined as in Remark 3. Now, we can obtain the values of $\rho^t \mid_{\mathcal{T} \times \mathcal{T}}$ for every $t \in [t_k, t_{k+1})$ by $M(k)(t) = \left( f_k^{(p_i, p_j)}(t) \right)_{i,j} \in \mathbb{R}^{|\mathcal{T}| \times |\mathcal{T}|}$.*

**Remark 12.** *There are some possible scenarios in which this condition is satisfied:*

- *The values of c are discrete, i.e., $t \mapsto c_a(t)$ is a step function for all $a \in A$. Then, $t \mapsto \rho^t(u,v)$ is a step function for all $u,v \in V$ (as V is finite). We can conclude that we can find a partition $[t_1, t_2) \cup [t_2, t_3) \cup \dots \cup [t_{n-1}, t_n) = I$ such that for every $k \in \{1, \dots, n-1\}$, we have $\rho^u = \rho^v$ for all $u, v \in [t_k, t_{k+1})$. Hence, we can define $f_k^{(u,v)}(t) := \rho^t(u,v)$ as constant functions for all $u, v \in V, t \in [t_k, t_{k+1})$.*
- *The class of polynomial splines is closed under composition and can be factorized after composition for optimized evaluations.*

Finally, we can state a variation of Problem 1 for the TDSP on time-dependent quasimetric networks.

**Problem 3** (TDSP Ranking Problem). *Let $G = (V, A, I, c)$ be a time-dependent quasimetric network, $\mathcal{T} \subset V$ be the set of targets, and $(s, t_s) \in V \times I$ be the query data. We want to know the relation $\leq_s^{t_s}$ given by the equivalence*

$$u \leq_s^{t_s} v \Leftrightarrow \rho^{t_s}(s, u) \leq \rho^{t_s}(s, v) \ \forall u, v \in \mathcal{T}.$$

**Solution 2.** *Let us assume the prerequisites of Problem 3 and let us additionally assume that G is discretizable with*

$$[t_1, t_2) \cup [t_2, t_3) \cup \dots \cup [t_{n-1}, t_n) = I$$

*so that we can compute the matrices $M(i)$ for all $i$. Then, we already have the information about $\rho^t(u,v)$ for all $u, v \in \mathcal{T}$ and each $t \in I$ using the matrices $i \mapsto M(i)$ for $i \in \{1, \dots, n\}$. Thus, we can modify Algorithm 1 to use the bounds (21) and (22), respectively, instead of (2) and (3) for a query vertex $s \in V$ and a query time $t_s \in I$. We can follow Solution 1 by lifting each step to the time-dependent case, e.g., setting $p := \text{argmin}_{v \in \mathcal{T}} \, d^{t_0}(s, v)$ for a heuristic algorithm and a given time-dependent lower bounding metric $d^t$.*

**Remark 13.** *The difference between Problem 1 and Problem 3 lies in the fact that we exchanged the classical triangle inequality with the time-dependent one. We have to take care when transferring Problem 2 to the time-dependent case, because the assumed equation*

$$\rho^t(u,v) + \rho^{t + \rho^t(u,v)}(v,u) = \rho^t(v,u) + \rho^{t + \rho^t(v,u)}(u,v)$$

*does not hold in general (i.e., a time-dependent round trip may be asymmetric).*

## 6. Evaluation

In what follows, we present an implementation of Solution 1 for Problem 1. Remember that we have a quasimetric network $G = (V, A, c)$ and a set of target vertices $\mathcal{T} \subset V$. In a preprocessing step, our solution builds the matrix $M$ (see Remark 3). Then, for a given query vertex $s \in V$, we compute the ranking of all vertices of $\mathcal{T}$ based on the distances from $s$ by running an SSSP algorithm and using the information stored in $M$.

In what follows, we present statistics on the quality of $M$ (Section 6.1), sketch algorithmic ideas of our implementation (Section 6.2), explain our implementation in detail (Section 6.3), and finally explain our experiments (Section 6.4).

### 6.1. Statistics

The success of our $\delta$-criterion (cf. Lemma 4) depends on the distribution of the targets and the distances between them. Dependent on the distribution of the targets, the number of targets can also have an impact. On strongly asymmetric graphs, the $\gamma$-criterion (cf. Lemma 6) can be a more helpful strategy to choose than the $\delta$-criterion, which emphasizes the round-trip distance to the closest target according to the Euclidean distance. For the statistical evaluation, we used the OSM data of San Francisco's bay area (see [1] for a description of the dataset). The vertices of the graph are enumerated so that we could randomly pick (in a uniformly distributed manner) some targets of this set. We observe that the minimum of distances between targets correlates with the mean value of $\delta$. Figure 3 shows that our uniform selection of targets induces an anti-correlation between the number of targets and $\delta$. Last but not least, if the set of targets spans an equally distant lattice, the draw area of the $\delta$-criterion is independent of $|\mathcal{T}|$.

From this evaluation, we can conclude that we can leverage the $\delta$-criterion only for special distributions of the target vertices. Its strength suffers from the uniformly random distribution of the chosen target vertices. However, we will later show that we can still perceive a speedup when caching results for subsequent queries.

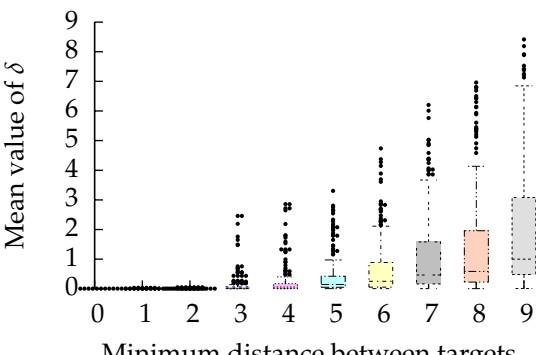

**(a)**

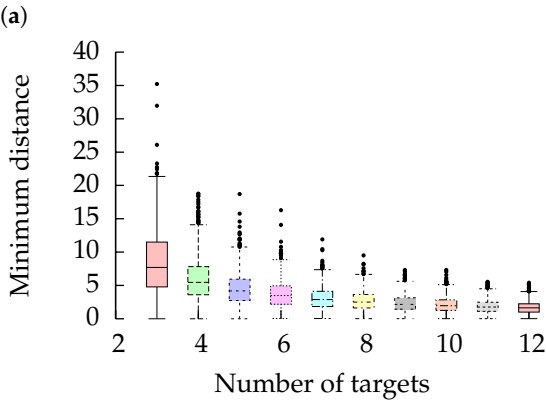

**(b)**

**Figure 3.** Statistical evaluation of the distances of the target vertices on the San Francisco dataset. (**a**) Correlation between $\delta$ and the minimum distance between all targets. (**b**) Anti-correlation between $|\mathcal{T}|$ and the minimum distance between all targets, caused by the uniform distribution.

### 6.2. Design and Overview

We have implemented Algorithm 1 as a deferred callback function. This function can be wrapped up with every common SSSP algorithm. For evaluation, we have implemented

a modification of Dijkstra and A*. The new algorithms take a list of targets and try to find the shortest paths to each target. Evaluation of A* with multiple targets becomes tricky. We can either

1. Choose a fixed function at the beginning of A*; or
2. Update the heuristic whenever another specific target vertex is found. Updating means reevaluating the heuristic estimate for each vertex that is currently in the open front. This takes time and shows that Dijkstra is faster in situations where the target vertices are arbitrarily distributed on the network.

We have chosen the second strategy, since it gives the callback function the necessary time to evaluate before the next target is found. Now, back to our modification. Furthermore, we let each SSSP algorithm call the callback function whenever it has found a shortest path to one of the targets. We implement Remark 6 by passing a list of already cached vertices to the shortest-path algorithm. Again, whenever a shortest path to one of those cached vertices is found, we have to check if we can solve the problem.

Without sophisticated partitioning of the graph [51], it is hard to parallelize an SSSP algorithm. Bidirectional search gives us at least the chance to let two threads run in parallel [52]. For simplicity, we stick to basic sequential versions but run a search for the distances $\rho(s, \mathcal{T})$ and $\rho(\mathcal{T}, s)$ in parallel for each routing query. Therefore, we do not follow the sequential concept of Algorithm 1. Instead, as shown in Figure 4, after invoking both forward and reverse SSSP algorithms, the main thread simply waits for termination of the forward algorithm. That is because we have modified both SSSP algorithms to evoke a callback function asynchronously that does the job of Algorithm 2. Thus, our modified SSSP algorithm will run just marginally slower than the standard version. After the forward search exits, we have to check whether the forward search has completed or was interrupted by a callback: We have either calculated the shortest path lengths to all targets, or we have gathered enough information to construct the ordering relation in question.

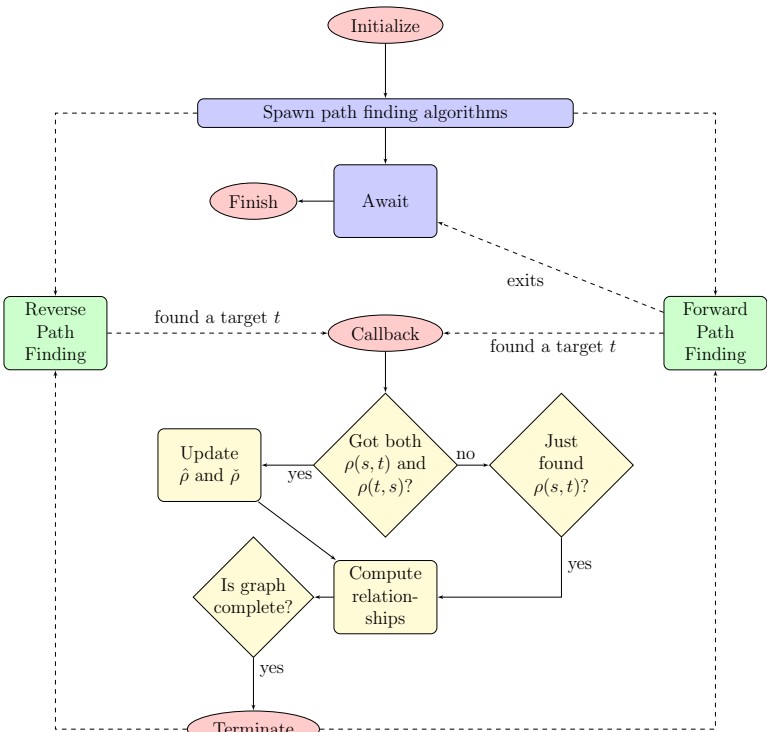

**Figure 4.** Flowchart of our asynchronous algorithm determining the ranks of the target vertices. The flowchart shows asynchronous (dashed arrows) and synchronous (solid arrows) execution of tasks (rounded rectangles). Tasks of different threads are grouped by color: The main thread is blue, both shortest-path-finding threads are colored in green and the callback thread is yellow.

### 6.3. Implementation

On modest hardware, large routing networks cannot be kept in memory. Hence, there is the need for a database to save and query network graphs. For our implementation, we stick to graph databases [53]. These databases lack concepts of relational databases such as joins or foreign keys, but they take advantage of many-to-many relationships [54] that inherently exist on routing networks. Our actual implementation utilizes Tinkerpop's Blueprint library (https://tinkerpop.apache.org/, accessed 19 December 2021), whose interface is based on a labeled multigraph [55]. Blueprint is an abstraction layer for graph databases such as Neo4j [56] or OrientDB [57]. It further has an in-memory implementation called TinkerGraph (https://tinkerpop.apache.org/javadocs/3.2.2/full/org/apache/tinkerpop/gremlin/tinkergraph/structure/TinkerGraph.html, accessed 19 December 2021) that also obeys the Blueprint interface. We use both TinkerGraph and the Neo4j overlay to benchmark our algorithm. As opposed to traditional Java programming, we have written the complete test suite in Clojure [58,59]. Clojure is a Lisp dialect that compiles to the Java virtual machine (JVM). It leverages the task of asynchronous programming by adding concepts of atoms, agents, and transactional references to its core language [60]. Clojure does not follow the common approach of pessimistic locking, i.e., to wrap thread-unsafe parts with locks: The language embraces the so-called Software Transactional Memory concept [61] for data manipulation. It follows the idea of the ACID concept [62] that is predominant in the field of relational databases but without data durability, because variables still remain transient in main memory. Whenever a transaction fails due to a race condition, it is reset and restarted. Unfortunately, thread-unsafe Java methods cannot be lifted to this transactional model. That is why we use a readers–writers lock for generating the ranking order; which by the way is realized by a Tinkerpop's TinkerGraph. If we reconsider our ideas about dynamic caching in Remark 6, we could represent the matrix $M$ as a list of arrays that grows with newly cached values. This matrix would get sparser the more data are cached. That is because newly cached vertices only need to know the distance (there and back) to every target but not to already cached vertices. To prevent waste of space, we realized the matrix $M$ as a hash-map with a pair of vertices as key type. For simplicity, the value will be cached whenever both forward and reverse search had visited all target vertices. A more sophisticated approach would take the evaluation time of both SSSP algorithms into consideration and store a vertex whenever an expected running time is exceeded. The cache can be purged if the space gets sparse. A more dynamic approach would employ a background task that frequently dismisses recently unutilized cached vertices.

### 6.4. Experimental Results

For the following evaluation, we generated the matrix $M$ in a preprocessing step for each given routing network. To avoid outliers in the time measurement, we took sufficiently many samples and computed the average time of those; we empirically evaluated that running 30 iterations was enough to compute a stable average time. Each shown execution time in the figures is the mean value of the measured times without both the smallest and largest outliers. All times are shown in milliseconds. Each *y*-axis with label *Targets founds (%)* measures the relative number of targets that are found by our modified SSSP algorithm before it could compute the ranks of all targets. The lower this percentage, the less work the SSSP algorithm had to do thanks to the gathered information.

To keep the figures small, we abbreviated Dijkstra to *Dij.* and prefix our modified SSSP algorithms with *imp.* (for our implementation).

The tests ran on a single node with an Intel i7-3770 CPU and 16 GB of RAM. The operating system is Debian 7.1. The technical prototype of our framework is available for re-evaluation or implementation at https://github.com/koeppl/idron (accessed at 19 December 2021). The implementation supports the direct import of OSM datasets. For instance, the later used routing network in the Munich area (Section 6.4.3) can be directly

### 6.4.1. Star Graph

We have seen in Section 6.1 that a uniform distribution will unlikely show good results
for evaluation. For the evaluation on synthetic data, we propose a model of a star-shaped
graph. The graph consists of $|\mathcal{T}|$ line graphs of $\ell$ vertexes, which are called *axes* of the
star-shaped graph; these are connected at a center vertex $s$, which is also the query vertex
in our setting. For each axis, we promote exactly one of its vertices to a target vertex. In
other terms, the underlying graph is a tree with $s$ as the root, and no path from $s$ to a
target vertex contains another target vertex. See also Figure 5 on the left for a sketch. The
idea behind this model is the following: Given a network $G$, a query vertex $s$, and a set
of target vertices $\mathcal{T}$, $G$ can be contracted to a star-shaped graph with $s$ as its center. We
can do this without changing the distance values $\rho(s, u)$ for each $u \in \mathcal{T}$ by tweaking the
edge costs adequately and thus without changing the ordering relation. In that sense, with
benchmarks on our star-shaped graph, we can infer the performance of our solution on
arbitrary routing networks.

For simplicity, all arcs are equidistant, i.e., $c(a) = 1 \; \forall a \in A$. We compare our imple-
mentation (imp.) with a naive approach that computes all shortest paths from $s$ to $\mathcal{T}$. The
tests of Figures 6 and 7 are evaluated on a star graph with variable axis lengths and $|\mathcal{T}| = 15$
axes/target vertices. There are $\ell \cdot i^2 / |\mathcal{T}|^2$ vertices between $s$ and a target vertex $u_i \in \mathcal{T}$ in
Figure 6, and $\ell \cdot i^2 / |\mathcal{T}|^2 + |\mathcal{T}|/2$ in Figure 7 (here, the axes have a length $\ell + |\mathcal{T}|/2$).

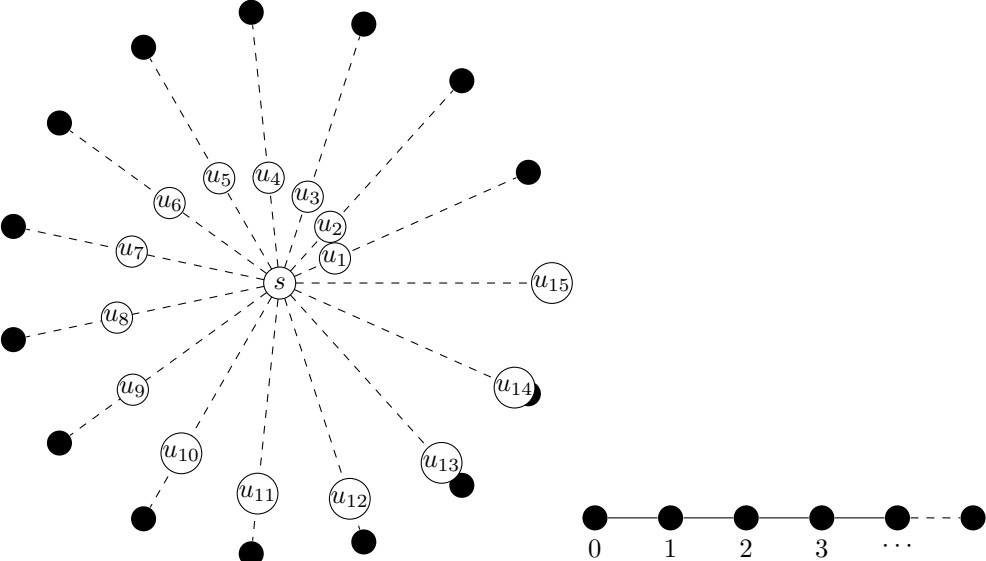

**Figure 5.** Synthetically generated graphs considered in Sections 6.4.1 and 6.4.2. Dashed lines visualize
simple paths. On the left, the query vertex $s$ is in the center, and $\mathcal{T} = \{u_1, \ldots, u_{15}\}$. On the right, we
assign each vertex an index by enumeration.

From the left plot of Figure 6, we observe that the execution times correlate with
the parameter $\ell$ for the naive implementation. However, for our implementation, the
distribution of the targets is of importance: the further away a target is from $s$, the larger
the distance is between its neighbors. Some targets are very close to $s$, which is an excellent
precondition for the $\delta$-criterion. In fact, from the left plot of Figure 6, we observe that
the algorithm is quite stable with variable graph sizes. In the right plot, we observe that
$\ell$ is anti-correlated with the number of target vertices visited by the SSSP algorithm. In
other terms, the larger $\ell$ gets, the less work the SSSP algorithm has to do—we can infer
the complete ranking based on the distances from $s$ to only a few target vertices for large
values of $\ell$.

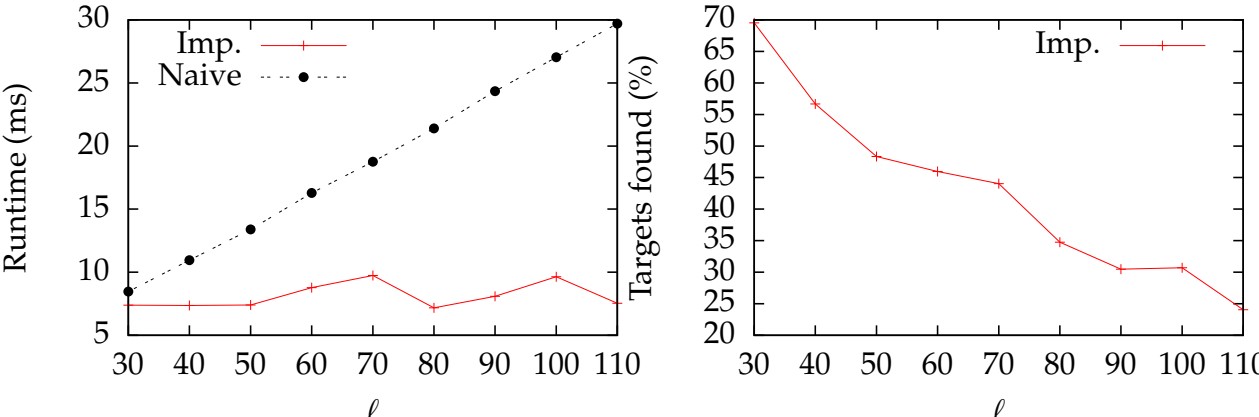

**Figure 6.** Evaluation of a star-shaped graph with the query vertex $s$ in the center. Each axis of length $\ell$ of the star is assigned exactly one target vertex $u_i$ whose distance to $s$ is $\ell \cdot i^2 / |\mathcal{T}|^2$. Since the arcs all have unit costs, the length between two nodes $(u, v)$ is the number of nodes of the shortest path from $u$ to $v$. *Naive* and *Imp.* are measuring Dijkstra's algorithm and Dijkstra's algorithm combined with our solution, respectively.

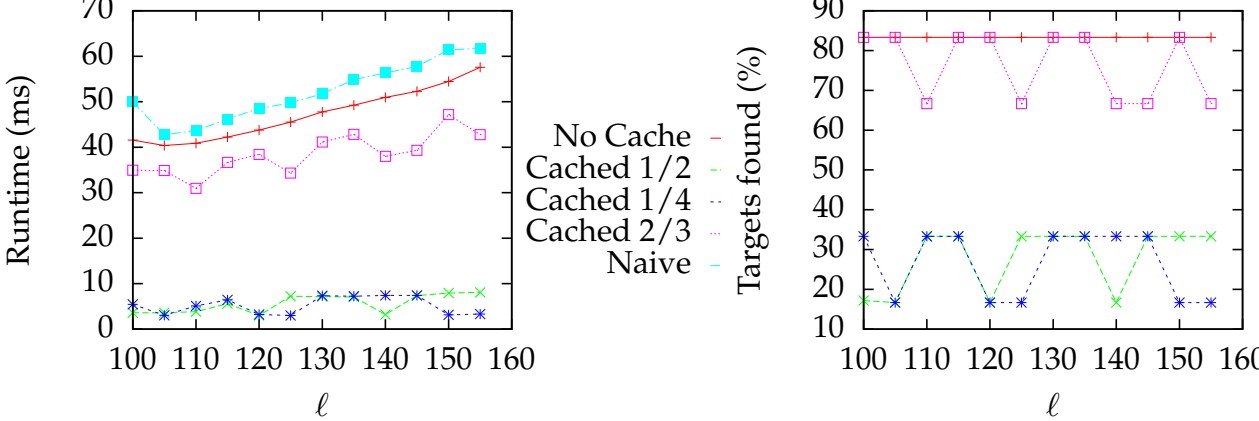

**Figure 7.** Evaluation of a star-shaped graph similar to Figure 6. Here, the distance between $s$ and a target vertex $u_i$ is $i^2 + |\mathcal{T}|/2$. We evaluated Dijkstra's algorithm (*Naive*), our implementation using Dijkstra's algorithm without caching (*No Cache*), and when caching a query vertex whose distance to $s$ is $1/4$, $1/2$, and $2/3$ of the distance from $s$ to $s$'s closest target. We call these instances *Cached 1/4*, *Cached 1/2*, and *Cached 2/3*, respectively, in the legend.

In the setting of Figure 7, there are no target vertices in the close vicinity of $s$. Hence, we have an offset that badly degenerates the performance, since the algorithm first has to find at least one target or cached vertex. The trick here is to cache the vertex $s$ so that a new query $s'$ in the close vicinity of $s$ can use the cache as an advantage. We have evaluated the algorithm once with a cleared cache (called *No Cache*) and with a cached vertex whose distance to $s$ is $1/4$, $1/2$, and $2/3$ of the distance from $s$ to the target closest to $s$. Due to the nature of the $\delta$-criterion, a ratio of $2/3$ seems not to be of value.

**Remark 14** (Outlier Target Vertices). *As with statistical outliers, there often is a target that is much farther away from $s$ than most targets. Without our approach, an SSSP algorithm has to figure out the complete path from $s$ to this vertex, just to determine that it is indeed far away. In applications where the user or system knows the area of interest, we can specify a clipping region in order to skip routing to these vertices. However, a clipping region cannot be constructed automatically. If it is too small, the answer can be unsatisfying because the desired results are*

*missing. If it is too large, the benefit is negligible. With our approach, path-finding stops when the complete ranking is constructed. Therefore, routing will stop before searching for outlier targets.*

### 6.4.2. Line Graph

In this evaluation, we take a simple path as a graph, called a line graph in the following. Figure 5 on the right sketches such a graph. We enumerate the vertices such that the leftmost vertex has the index 0 and the last vertex has an index equal to $|V| - 1$. While we fixed in the prior experiment (Section 6.4.1) the query vertex, we promote successively vertices with increasing index values to the query vertex, and we perform a query, keeping $\mathcal{T}$ fixed.

In the settings for Figure 8, we promote the start and the end of this line graph to target vertices $u$ and $v$. We observe a peak in the running time in the middle of the way between both targets. For that query vertex, both targets have the same distance $\rho(s, u) = \rho(s, v)$. Without caching (Curve *No Cache*), our algorithm has to do a complete path search. If we allow the algorithm to cache this query vertex (Curve *With Cache*), we observe a drop of the execution time instead of the peak.

In the next experiment, we increased the number of target vertices, which are now equidistantly placed along the line graph. In Figure 9, we observe a wave-like curve for the performance. The algorithm performs well when the query vertex is close to one of the targets. On this type of graph and type of distribution, Dijkstra is faster than A*. Despite that, our algorithm has nearly equal running times with both Dijkstra and A*. The right plot of Figure 9 leads us to the conclusion that the implementation with A* does not access as many vertices as the implementation with Dijkstra, and therefore, it benefits more from our preprocessing than Dijkstra.

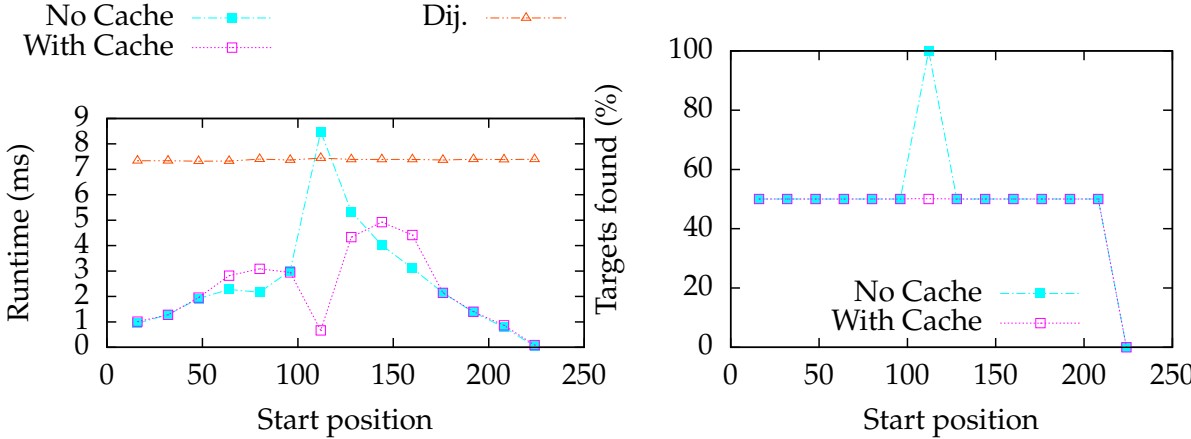

**Figure 8.** Evaluation of a line graph with two target vertices $u$ and $v$ having the indices 0 and 250, respectively. We promote vertices with an index from 10 to 225 to the query vertex $s$; the $x$-axis reflects this index (i.e., the start position is the index of the start vertex). *Naive* measures the time for Dijkstra's algorithm to find the shortest paths to all target vertices. The curves *No Cache* and *With Cache* measure our solution when running the computation without cache and running the computation with the node in the middle of the line graph cached, respectively.

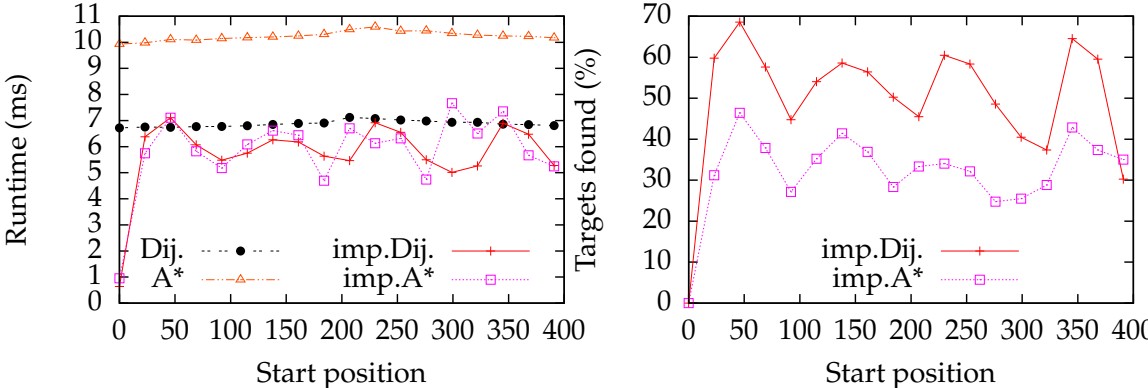

**Figure 9.** Evaluation of a line graph with $|\mathcal{T}| = 13$ equidistantly distributed target vertices; the vertex with index 0 is a target vertex. As in Figure 8, we successively promote vertices with an index from 0 to 400 to a query vertex (this index is reflected by the *x*-axis, i.e., the start position), and run a query. We do not promote any target vertex to a start vertex, except for index 0. The curves *Dij.* and *A*\* measure the time for Dijkstra's algorithm and the A\* algorithm for finding the shortest paths to all target vertices. The curves *imp. Dij.* and *imp. A*\* measure our implementation using Dijkstra's algorithm and the A\* algorithm, respectively.

### 6.4.3. Show Case: City of Munich

Finally, we imported a small part of Munich (the area surrounding the central station) from OSM into a Neo4j database. See Figure 10 for a visualization of the OSM routing network. We used seven target vertices and only query vertices that are connected with every target. We evaluated our algorithm over three rounds; a round consists of batch computing the solution for a fixed list of query vertices. In the first two rounds, the cached data of roughly 20 query vertices are kept. Round 2 follows Round 1 consecutively with the same list of query vertices. The cache was pruned and kept empty for the last round (i.e., discarding caching). For the plot in Figure 11, we used Dijkstra as our SSSP algorithm.

Since the chosen area is relatively small, we cannot see a great variation in the runtime costs for classic Dijkstra. Our implementation using no caches has arbitrary performance: On the one hand, for well-chosen query vertices, it is significantly faster than the classic Dijkstra algorithm. On the other hand, for bad choices, it sometimes performs even slightly worse. Fortunately, caching significantly improves the performance: We observe that in both rounds, our algorithm performs always better than Dijkstra, and most of the time, it performs significantly better. The first roughly 20 start positions in the plot correspond to cached results such that the query can be instantaneously solved by consulting the cache. For Round 2, we see negative spikes (for instance, at *Start position* $\approx$ 100), where a cached result was used that was not cached during Round 1 when querying that vertex.

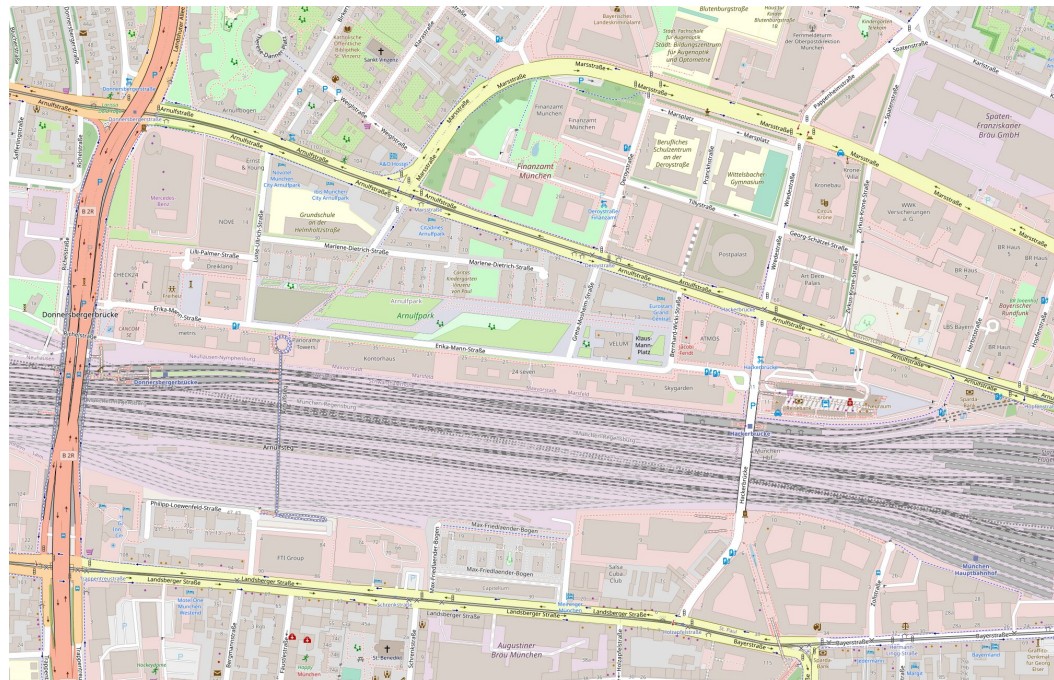

**Figure 10.** Visualization of the Munich dataset via OpenStreetMap, covering the indexed area [48.14, 48.145] north latitude and [11.54, 11.543] east longitude from www.openstreetmap.org (accessed on 19 December 2021) with a scale factor of 16. The map is oriented north-up. © OpenStreetMap contributors.

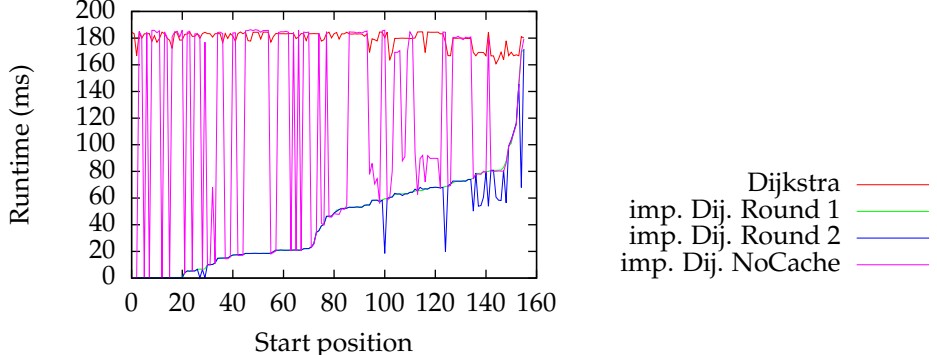

**Figure 11.** Random queries on the Munich dataset. The start vertices are enumerated by start positions (the *x*-axis in the plot), which are ranked ascendingly with respect to the time consumed by our algorithm in its first round. The curves *Dijkstra* measure the time for Dijkstra's algorithm to find the shortest paths to all target vertices. The other curves with prefixes *imp. Dij.* denote our solution using Dijkstra's algorithm. The implementation with suffix *NoCache* does not cache past queries, while *Round 1* caches query vertices, and *Round 2* makes use of the cache of *Round 1*.

## 7. Discussion

We conclude our experiments in Section 6.4 with a discussion of its results. The discussion is divided into the experiments of that section.

### 7.1. Star Graph

For target vertices spaced in the distribution of Figure 6 in Section 6.4.1, our solution runs in empirically constant time, since there are enough target vertices in constant distance to *s* such that an SSSP algorithm such as A* can find them in constant time, serving enough information to infer the complete ranking. If the target vertices are far away from *s* and are not necessarily well distanced, our preprocessing is of less use, but it is still faster than the

naive algorithm. We can obtain a remarkable speed-up when caching a vertex close to *s*. This is especially useful for use cases with consecutive queries.

### 7.2. Line Graph

Performance deteriorates when the distances from the query vertex to the target vertices are close. By leveraging caching, we can successfully speed up the queries in such cases such as in Figure 8 of Section 6.4.2. Although A* is faster than Dijkstra on graphs where it can leverage its heuristic, its performance deteriorates on types of graphs such as in Figure 9, but its modification can perform as fast as our modified Dijkstra. We conclude that our modification makes A* a more robust approach.

### 7.3. Show Case

As observed in Figure 11 of Section 6.4.3, without caching, our solution is very sensitive with respect to the query vertex. A bad choice makes it not visibly better than a plain SSSP algorithm. However, by leveraging caching, our algorithm can improve its performance significantly by storing information crucial for answering a subsequent query in the vicinity of past query vertices.

## 8. Conclusions and Future Work

In this article, we have studied the problem of ranking a set of target vertices with respect to the distance from a start vertex on a directed weighted graph. We considered the graph as well as the target vertices as static, supporting preprocessing of the graph with respect to the target vertices to accelerate subsequent queries.

### 8.1. Theoretical Conclusions

Compared to the classic Single Source Shortest Path Problem (SSSP), a ranking of the target vertices involves new aspects of optimization because the exact calculation of distances is not required for answering ranking queries of this type. We have seen that a database that answers this type of query on a static dataset can benefit from preprocessing distances between all vertices in question. On the one hand, the preprocessing of our method is not only much faster than of the ALT algorithm or other preprocessing approaches such as hub labels and contraction hierarchies. We also used only marginal space for providing a matrix of distances with $\mathcal{O}(|\mathcal{T}|^2)$ space, instead of $\mathcal{O}(|V||\mathcal{T}|)$. On the other hand, we cannot guarantee that we can determine the ranking order in one step with our negligible preprocessing. Instead, we might have to visit some targets before we can determine the ranking. We have done this by defining lower and upper bounds for the routing network distance, which we could improve while gaining more and more knowledge about the shortest paths from the given query vertex. With the $\delta$-criterion, we obtained an a priori estimation, and with the $\gamma$-criterion, we obtained an interim estimation of areas. Within the proposed areas of both criteria, we can be certain about the ranking. Additionally, this gave us the opportunity to compute validity regions in which the query vertex can move freely without changing the result. As a last step, we have transferred our posed problem to time-dependent routing networks and found similar results.

### 8.2. Practical Conclusions

We have implemented our algorithm computing the ranking of the target vertices in terms of an asynchronous application framework in such a way that it is easy to embed it into already existing shortest-path algorithms, ranging from simple applications of Dijkstra's algorithm up to complex shortest-path computation with sophisticated preprocessing techniques such as contraction hierarchies. To keep the analysis of the net benefit simple, we compared our solution with a simple application of Dijkstra's algorithm or the A* algorithm on the plain graph without any preprocessing techniques. In this setting, we observed that the benefits of our solution largely depend on the distribution of the target vertices and the choice of the start vertex. For bad choices, no benefits could be observed. As a remedy, we

proposed to cache the results of slowly answered queries. Enabling caching, we observed that subsequent queries within the close vicinity of a cached query vertex boosted the query speed considerably. Therefore, we can conclude that our proposed indexing data structure is especially useful for settings where multiple queries are issued within the vicinity.

*8.3. Future Work*

The current evaluation only compares the presented techniques with a standard Dijkstra or A* implementation without preprocessing. Further investigations are required to judge where and to what extent our presented techniques can be implemented in existing systems using preprocessing techniques for shortest-path queries. Therefore, we want to evaluate our prototype with state-of-the-art preprocessing techniques such as the approaches using customizable contraction hierarchies. We are positive that a combination of those techniques with the techniques presented here could lead to an improved solution for our addressed problem of determining the ranking of the target vertices. Moreover, we addressed the time-dependent routing only theoretically. We strive to enhance our implemented prototype for the use case of temporal routing networks, where we want to conduct a practical evaluation and comparison with known solutions on that type of networks.

On the theoretical side, we want to investigate whether the lower bound of $\Omega(\sqrt{|V|})$ on the average number of vertex visits set by Rupp and Funke [27] can be transferred to our problem when augmenting contraction hierarchies or hub labels with our indexing data structure.

**Funding:** This work is funded by the JSPS KAKENHI Grant Numbers `JP21K17701` and `JP21H05847`.

**Institutional Review Board Statement:** Not applicable.

**Informed Consent Statement:** Not applicable.

**Data Availability Statement:** Not applicable.

**Acknowledgments:** We thank Roland Glück for comments and discussion on this topic.

**Conflicts of Interest:** The author declares no conflict of interest.

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
