# Peer review of "Inferring Spatial Distance Rankings with Partial Knowledge on Routing Networks"

_information, doi:10.3390/info13040168_

Round 1
Reviewer 1 Report
The method that is described in the article is quite interesting and provide a more effective solution than the Dijkstra or A*. The main problem is the outdating of the paper. It seems the method was designed in 2014. The last reference is from 2013. On GitHub, the last commit is in 2014. After a quick googling, I found a lot of methods that deal with this problem and looks interesting and provide nice results.
The following part of the paper must be improved before publishing:
- the literature review must be completed (It means to consider the papers published after 2014)
- the proposed method should be compared to the state-of-the-art methods (It means to consider the methods developed after 2014). Implement the concurrent methods and compare the speed and precision (the proposed algorithm is not exact).
Author Response
Dear Reviewer 1,
Thank you for your suggestions.
I reviewed the state-of-the-art methods regarding the preprocessing on routing networks,
and in particular those that are time-dependent, i.e., work on temporal routing networks.
Despite the fact that the presented work seems outdated, the problem addressed in this article should be (still) novel.
Also, I think that the ideas are orthogonal to existing approaches, meaning that it is possible to enhance existing routing algorithms or
preprocessed routing graphs with our proposed method to speed up the execution.
Nevertheless, this would involve elaborately empirical testing,
and I would therefore suggest to consider this as future work.
To this end, I have described the plan for an in-depth implementation, evaluation, and comparison of those methods in the future work section at the end of the manuscript.
Reviewer 2 Report
Review: Inferring Spatial Distance Rankings with Partial Knowledge on Routing Networks
Manuscript ID: information-1540261
Journal: MDPI Information
Review Text:
The papaer describes an interesting problem in the field of routing. The authors highlight a solution to rank points with respect to their distance from a source - based on a pre-computed partial solution. The paper is written in an understandable language, and has a good structure.
The abstract is way too short and does not reveal a motivation and any results (or at least tells the reader what to expect in terms of a scientific contribution).
In section 1 I would not suggest to use "beelines" to denote the euclidean distance. In addition, GPS-enabled is misleading as most mobile devices use a multitude of positioning systems (e.g. Galileo, Glonass, Beidou, ...). Hence, I would suggest to use the term Global Navigation Satellite System (GNSS)-enabled.
Please use scientific language - e.g. "Deng et al. [8] share with us very similar thoughts" is rather non-scientific language.
Section 2 is clearly described and justified. Well done. Section 3 starts with well-known definitions - which are mentioned for the sake of clarity. Section 3 and section 4 are mathematically well described.
The authors could emphasize the results even more. As of now, it is not crystal clear if the proposed solution is better than A* and/or Dijkstra. The text, especially in the results section is not easily understandable and straightforward. I would strongly suggest to revise the results section, and maybe add a section "discussion of the results" in order to make the results more explicit.
Generally, an interesting paper with some weaknesses in the presentation.
Reviewer 3 Report
This manuscript introduced a novel method of inferring distance rankings by precomputation. The method is described in sufficient detail, and the authors have proven the effectiveness of the proposed method on both static and time-dependent networks. Overall, this is an interesting work and the proposed method has the potential to be applied in location-based services at scale. I would suggest adding some descriptions of the synthetic dataset and the OSM datasets.
Author Response
Dear Reviewer 3,
Thank you for your suggestion.
I have added descriptions from where I obtained the OSM datasets, and how to import the datasets into the experimental framework.
The synthetic datasets are now described more in detail. I also added sketches of their shapes.
Reviewer 4 Report
This manuscript submitted by the author cannot be reviewed by the reviewer due to not having the corresponding format of the Journal. Therefore, this article is rejected for failing to comply with this regulation, for which the author is requested to review the following web page: https://www.mdpi.com/authors
Author Response
I am sorry to hear that the formatting is indeed a requirement.
Prior to submission, I got positive feedback from Ms. Yilia Mao of the MDPI Information editorial office about whether submitting an unformatted draft as the first version is acceptable.
Nevertheless, the manuscript is now formatted in the document class provided by MDPI.
I would be glad to receive your feedback.
Round 2
Reviewer 1 Report
I must reject the article. That areas must be improved:
- The literature review should be improved. It is still incomplete. I recommend starting here: https://arxiv.org/pdf/1504.05140.pdf?ref=https://githubhelp.com
- Make a comparison (query time, preprocessing time, correctness) with other relevant methods. It seems that there are a lot of methods that provide a much better result (I am not sure about preprocessing time). I can not find any information about preprocessing time in the article draft. You should write in which aspect is your algorithm better than concurrence algorithms.
Author Response
Thank you for your recommendations.
I am in particular thankful for the link to the survey "Route Planning in Transportation Networks".
The explained approaches within this reference are quite exhaustive and well explained.
Together with some other references, I could enhance the related work.
I have also added an explanation of why the experiments are done without using any pre-computation technique, like those addressed in this survey.
The intension is that the ideas addressed in this manuscript can be used on top of any shortest path computation algorithm.
This also means that we can use state-of-the-art techniques like hub labels or contraction hierarchies as
a precomputation step to speed up the pre-computation as well as the actual computation of the proposed approach.
However, such an empirical analysis would go beyond the current scope, and thus I would like to address this analysis as future work.
Reviewer 2 Report
Dear authors,
thanks for the revised manuscript. The authors revised the manuscript according to the suggestions of the reviewers.
Hence, the paper is ready for publishing.
Author Response
Thank you so much for reviewing this manuscript and your helpful comments!
Reviewer 4 Report
Thanks to the author for performing the changes suggested by the reviewers. However, author still continue with an old MDPI format, so before publishing, these observations must be performed:
-In the article, it appears as a single author of the manuscript, so in the text of the article the 3rd person must not be used, because it does not include more authors.
-The verb tenses in the Sections are incorrect.
-There are even acronyms whose meaning is not correctly written.
-In the structure of the article must be all the Sections, and not the Subsections. And where is the Conclusion Section?
-The author must separate the discussions in a Discussions Section before the Conclusions Section.
-The authors have not placed any Figure on the scenarios, they have used from the OSM, from the city of Munich.
-The authors have not verified the proposal in any simulator.
-The authors do not disclose the parameters they have used to apply the proposal in the case study.
Round 3
Reviewer 4 Report
Thanks to the authors for performing the changes suggested by the reviewers. However, I will comment on some aspects that must be fixed before the publication of this manuscript:
-There are acronyms that are incorrectly misspelled. The correct form to write is to write the meaning with the initial letters in capital letters, as in line 17.
-The format of the submitted manuscript is the old one, so the author must verify the new format.
-The “Related Work” Subsection is a Section and not a Subsection.
-Subsection 1.2, 1.3, must be included in the Related Works Section.
-The author must write the manuscript only with font color black.
-The author must highlight only the changes performed, because in the manuscript is highlighted in gray and it confuses the reviewers.
-In line 219, there is a square.
-Figure 1 and 2 are not referenced in the manuscript.
-Algorithms are not being referenced in the text of the manuscript.
-Figure 3 is blurred.
-What is the reason for placing the “Distance Ranking Problem” in a box on line 123? That box must be removed.
-Footnote on page 17 must be deleted.
-Why the algorithm was executed 30 times?
-All the Figures must have their legend well defined and with clear names to have a better understanding.
-Figure 5 must be larger to observe better.
-The author must differentiate between a Section and a Subsection.
-Figure 7, 8, 9 and 11 do not clearly observe the lines in the legend.
-In Figures 6 and 7, on the ordinate axis it is length but it is not known in what unit it is.
-Eliminate the links that have been written in the manuscript.
-In Figure 8, 9 and 11, it is not understood exactly what the author refers to with “Start position”, it must be well explained both in the Figures and in the corresponding text.
-In line 513, “Figure 7” is repeated.
-In Line 540, “Figure 8” is repeated.
-There must not be a discussion in each experimental result that the author has placed.
-Figure 10 requires elements such as the coordinates of the maps on the axes, the scale it has and where the image was obtained from.
-In Line 575, the Discussion is a Section.
-The Conclusions and Future Work must be inside the Conclusion Section.
-The conclusions must be improved.
-There must not be a list of acronyms because they are written in the same text of the manuscript.
-Minority check the English language in the manuscript.
Round 4
Reviewer 4 Report
Many thanks to the author for making the suggested changes to the manuscript. However, only a thing that author must be done before publication is a minor revision to the English language of the article.